# *Rheum rhaponticum* and *Rheum rhabarbarum* Extracts as Modulators of Endothelial Cell Inflammatory Response

**DOI:** 10.3390/nu15040949

**Published:** 2023-02-14

**Authors:** Oleksandra Liudvytska, Michał B. Ponczek, Oskar Ciesielski, Justyna Krzyżanowska-Kowalczyk, Mariusz Kowalczyk, Aneta Balcerczyk, Joanna Kolodziejczyk-Czepas

**Affiliations:** 1Department of General Biochemistry, Faculty of Biology and Environmental Protection, University of Lodz, 90-236 Lodz, Poland; 2Department of Sociobiology and Epigenetics, Faculty of Biology and Environmental Protection, University of Lodz, 90-236 Lodz, Poland; 3The Bio-Med-Chem Doctoral School, University of Lodz and Lodz Institutes of the Polish Academy of Sciences, University of Lodz, Banacha 12/16, 90-237 Lodz, Poland; 4Department of Biochemistry and Crop Quality, Institute of Soil Science and Plant Cultivation, State Research Institute, Czartoryskich 8, 24-100 Puławy, Poland

**Keywords:** *Rheum rhaponticum*, *Rheum rhabarbarum*, rhapontigenin, rhaponticin, anti-inflammatory, cyclooxygenase, lipoxygenase, HR-QTOF/MS, metabolite profiling, molecular docking

## Abstract

Background: Inflammation, endothelial dysfunction, and alterations in blood physiology are key factors contributing to atherosclerosis and other cardiovascular disorders. Hence, modulation of endothelial function and reducing its pro-inflammatory and pro-thrombotic activity is considered one of the most important cardioprotective strategies. This study aimed to evaluate the anti-inflammatory potential of rhubarb extracts isolated from petioles and underground organs of *Rheum rhabarbarum* L. (garden rhubarb) and *R. rhaponticum* L. (rhapontic rhubarb) as well as two stilbenoids, typically found in these plants, i.e., rhapontigenin (RHPG) and its glycoside, rhaponticin (RHPT). Methods: Analysis of the anti-inflammatory effects of the indicated rhubarb-derived substances involved different aspects of the endothelial cells’ (HUVECs) response: release of the inflammatory mediators; cyclooxygenase (COX-2) and 5-lipoxygenase (5-LOX) expression as well as the recruitment of leukocytes to the activated HUVECs. The ability of the rhubarb-derived extracts to inhibit COX-2 and 5-LOX activities was examined as well. The study was supplemented with the in silico analysis of major components of the analyzed extracts’ interactions with COX-2 and 5-LOX. Results: The obtained results indicated that the examined plant extracts and stilbenes possess anti-inflammatory properties and influence the inflammatory response of endothelial cells. Biochemical and in silico tests revealed significant inhibition of COX-2, with special importance of rhaponticin, as a compound abundant in both plant species. In addition to the reduction in COX-2 gene expression and enzyme activity, a decrease in the cytokine level and leukocyte influx was observed. Biochemical tests and computational analyses indicate that some components of rhubarb extracts may act as COX-2 inhibitors, with marginal inhibitory effect on 5-LOX.

## 1. Introduction

Recent decades have been characterized by an increasing interest in the role of plant-derived substances in maintaining human health and studies on the physiological effects of phytochemicals occurring in foods and herbal medicines. For example, it has been found that natural, plant-derived compounds, such as resveratrol, curcumin, or epigallocatechin gallate (EGCG), can be helpful in the prophylaxis or treatment of many disorders, including cardiovascular diseases [1,2,3].

*Rheum rhaponticum* L. (rhapontic rhubarb, Siberian rhubarb) and *Rheum rhabarbarum* L. (garden rhubarb) belong to the *Rheum* L. genus (Polygonaceae), which includes about 60 herbaceous plants, native to the mountainous and desert regions of the Tibetan Plateau in China, and also growing in Asia and North America [4,5]. Today, rhubarbs are also commonly cultivated in Europe and North America [6] and used as food, medicinal, ornamental, and honey plants. While the edible stalks (petioles) of *R. rhaponticum* and *R. rhabarbarum* are typically known as foods, the roots (rhizome) of these species have been used for centuries in ethnomedicine of different regions of the world [4]. Both ethnomedicinal surveys and research data indicate that rhubarbs are rich in bioactive substances and may display beneficial effects on human health [7]. Numerous rhubarb products are on the market as foods, dietary supplements, and alternative medicines. However, only the phytoestrogenic activity of *R. rhaponticum* has been well evidenced, including basic research, animal studies and clinical trials [8,9,10,11]. Other physiologically relevant activities (including the impact on components of the cardiovascular system) of this and other rhubarb species are only partly or even poorly recognized. Moreover, some information on their properties is based only on traditional medicine recommendations. Thus, the biological activity of these plants needs to be examined.

The presented work includes the phytochemical profiling and evaluation of anti-inflammatory effects of extracts originating from petioles and underground parts of two rhubarb species, i.e., *R. rhabarbarum* and *R. rhaponticum* as well as two stilbenes, typically found in these plants—rhapontigenin (RHPG) and its glycoside, rhaponticin (RHPT). Experiments aimed to recognize their effects on the physiology of the endothelium and the possible protective role of these plant-derived substances on the inflammatory response of endothelial cells. The vascular endothelium is actively involved in inflammatory processes and the development of different cardiovascular disorders, e.g., atherosclerosis and thromboembolic complications. Pro-inflammatory factors released by endothelial cells may promote and augment both local and systemic inflammation [12,13,14]. Therefore, this study evaluated the anti-inflammatory effects of the examined extracts and compounds using an experimental model of human umbilical vein endothelial cells (HUVECs). Anti-inflammatory activities of the examined substances were analyzed at different levels of cell response, including the expression of cyclooxygenase (COX-2) and 5-lipoxygenase (5-LOX) genes, pro-inflammatory cytokine release and changes in the endothelium adhesiveness to leukocytes. Studies on HUVECs were enriched with COX-2 and 5-LOX screening tests and mechanistic in silico analyses of interactions of significant components of the examined extracts with the COX-2 and 5-LOX enzymes.

## 2. Materials and Methods

### 2.1. Chemicals

Commercial standards of derivatives of anthraquinones (aloe-emodin, aloe-emodin-8-Glu, emodin, emodin-8-Glu, rhein, rhein-8-Glu, chrysophanol, chrysophanol-8-Glu, physcion, sennoside-A, sennoside-B, sennoside-C, sennoside-D), stilbenes (resveratrol, pterostilbene, pinostilbene, piceatannol, astringin, polydatin, rhapontigenin, rhaponticin, isorhapontigenin, isorhaponticin, deoxyrhapontigenin, deoxyrhaponticin), phenolic acids (glucogallin, gallic-acid), flavones (vicenin-II, vicenin-III, apigenin-7-Glu), flavanones (pinocembrin, pinocembroside), chalcones (phloretin, phloridzin, trilobatin), and catechins (catechin, epicatechin, gallocatechin, procyanidin-B1, procyanidin-B2, procyanidin-B3, and procyanidin-C1), acetonitrile LC-MS grade, formic acid MS-grade, and *tert*-butanol were purchased from Merck (Darmstadt, Germany). Methanol, *n*-hexane, and *n*-butanol, all of analytical grade, were purchased from Fisher Chemical (Loughborough, UK). Ultrapure water was prepared using a Milli-Q water purification system (MerckMillipore). General reagents for bioassays were purchased from Sigma-Aldrich (a part of Merck KGaA, Darmstadt, Germany) and Cayman Chemicals (Ann Arbor, MI, USA). Reagents specific to cell culture, gene expression analyses, and cytokine profiling have been indicated below in the descriptions of the applied methods.

### 2.2. Plant Material

Petioles and rhizomes of *Rheum rhaponticum* L. were donated from The Botanical Garden of Maria Curie-Skłodowska University in Lublin, whereas analogous organs of *Rheum rhabarbarum* L. were given by The Botanical Garden of Łódź. The voucher samples have been deposited at the Department of Biochemistry and Crop Quality of the Institute. 

#### 2.2.1. Extraction Procedure

The plant material was cut into small, approx. 1 cm long pieces and dried in a dryer at 35 °C. The dried material was pulverized using an electric grinder to a homogeneous size and sieved through a 0.5 mm sieve. Powdered plant material was transferred into tightly closed vials and kept in the fridge until the analysis.

The plant material was extracted twice with methanol containing 0.1% formic acid, using an ultrasonic bath at room temperature for 24 h in the dark. The crude methanol extract was defatted using liquid–liquid extraction with *n*-hexane. Afterward, the specific metabolite fraction was prepared using *n*-butanol as an extraction solvent. All collected fractions (hexane, water, and butanol) were monitored for specific metabolites by the UHPLC-HR-MS method. Butanol fractions of petioles and roots were freeze-dried and used in the following research stages, LC-MS analyses and bioactivity assays.

#### 2.2.2. High-Resolution LC-MS Qualitative and Quantitative Analysis

The freeze-dried samples of rhubarbs’ butanol extracts (10 mg) were precisely weighed (with an accuracy of 0.1 mg) and dissolved in 80% (*v*/*v*) methanol in 1 mL volumetric flasks. Before analysis, samples were diluted 1:9 with 80% methanol and filtered using Whatman (0.22 µm) filter vials.

High-resolution UHPLC-MS analyses were performed on a Thermo Ultimate 3000 RS chromatographic system coupled/hyphenated with a Bruker Impact II HD (Bruker, Billerica, MA, USA) quadrupole time-of-flight (Q-TOF) mass spectrometer. Chromatographic separations were carried out on a Waters CORTEX T3 column (150 mm × 2.1 mm, 2.7 µm, Milford, MA USA) equipped with a pre-column. The mobile phase A was 0.1% (*v*/*v*) formic acid, and the mobile phase B was acetonitrile containing 0.1% (*v*/*v*) of formic acid. At the beginning of separation, the elution profile was isocratic, 5% of phase B for 0.5 min followed by a linear gradient from 5 to 98% of phase B in 18.5 min and a hold of 98% of phase B for 3 min. After completion of the separation, the column was equilibrated for 5 min with 5% of phase B. The separations were carried out at 35 °C with a constant flow rate of 0.5 mL/min. An injection volume of 5 µL was used.

The flow from the column was split between the charged aerosol detector (Thermo Corona Veo RS) and the ion source of the mass spectrometer type QTOF (Bruker Impact II HD) in a 3-to-1 ratio. This source, designed for electrospray ionization, operated with the following parameters: voltage capillary 2.8 kV; nebulizer pressure 0.7 bar; drying gas flow 6 l/min; drying gas temperature 200 °C; ion energy 4 eV; RF collision cell 700.0 Vpp; transfer time 100.0 μs; and pre-pulse storage 10.0 μs. Negative ions were measured in the *m/z* range of 100–1500 with a 5 Hz scanning frequency. MS/MS spectra were obtained in the data-dependent mode, in which two of the most intense ions were fragmented by collision-induced dissociation (CID, Ar collision gas). The collision energy and the ion isolation width for each fragmentation were automatically selected from a predefined list based on the *m*/*z* of the precursor ion. Internal mass calibration for quadrupole and TOF analyzers were based on sodium formate clusters automatically injected in a 10 mM solution in 50% 2-propanol into the ion source immediately before each analysis.

Complete absolute quantitation of all analytes was impossible due to the high diversity of observed metabolites and the lack of appropriate reference standards for most of them. Therefore, wherever possible, we semi-quantified the observed analytes based on the signal from CAD. The dependence of the signal intensity on the mobile phase composition was established using 32 reference standards representing metabolites present in the extracts. Each standard was analyzed at eight concentration levels (from 0.7 to 70 µg/mL) to obtain 3D calibration curves as proposed by [15]. Based on the concentration–response relationship established from this data, a linear universal response model linking retention time and detector response was established and used to provide semi-quantitative data on metabolite contents. 

The combined petiole and root samples spiked with 32 reference standards at two concentration levels (4 and 40 µg/mL) were used as quality control samples and analyzed after each block of 10 injections of regular samples. The Bruker Data Analysis, version 4.4 SR1, was used for data analysis and processing. Preliminary identification of rhubarbs’ metabolites was performed using high-resolution mass-to-charge (*m*/*z*) measurements, with errors not exceeding 5 ppm. Based on these results, summary chemical formulas were calculated. The agreement of these formulas with structural insights gained from MS/MS spectra was validated with computational methods using SIRIUS ver. 4.5.1 software [16].

Identification of detected compounds was mainly based on comparisons with reference spectra and MS/MS spectra interpretation aided by SIRIUS and MetFrag [17]. 

Each tentatively identified metabolite was assigned an identification confidence level class, as suggested by [18]. The class 1 metabolites were confirmed using authentic reference standards, whereas class 4 compounds were only assigned elemental sum formula and had no identifiable structural features in their MS/MS spectra.

### 2.3. Cell Culture

HUVECs (human umbilical vein endothelial cells) were cultured in MCDB-131 medium (Life Technologies, Carlsbad, CA, USA), supplemented with 10% heat-inactivated fetal bovine serum (FBS) (Life Technologies, Carlsbad, CA, USA), 10 ng/mL of epidermal growth factor (EGF) (Millipore, Burlington, MA, USA), and 10 mM glutamine (Invitrogen; Carlsbad, CA, USA). The cells were isolated from freshly collected umbilical cords, by collagenase type II digestion, according to Jaffe’s protocol [19], and used for the experiments at passages 3-4. The Bioethics Commission at the University of Lodz approved the study protocol (decision No. 15/KBBN-UŁ/III/2019 and 16 (III)/KBBN-UŁ/I/2021-22).

U-937 (ATCC, No. CRL-3253), a human monocyte cell line was cultured in RPMI-1640 medium (Promega, Madison, WI, USA) with 10% heat-inactivated FBS (Life Technologies, Carlsbad, CA, USA) according to a standard suspension cell culture protocol. For stimulation of the inflammatory response of the HUVECs, lipopolysaccharide (LPS) from *Escherichia coli* O55:B5 (Cas No. L4524, Sigma-Aldrich, St. Louis, MI, USA) was used. Indomethacin (CAS No. 53-86-1, Sigma-Aldrich, St. Louis, MI, USA) is a synthetic nonsteroidal indole derivative with anti-inflammatory activity (through cyclooxygenase inhibition). Zileuton (CAS No. 111406-87-2, Cayman Chemical, Ann Arbor, MI, USA) is a leukotriene synthesis inhibitor (5-LOX inhibitor). 

### 2.4. Effects of the Examined Substances on HUVEC Viability (Resazurin-Based Assay)

Cells were seeded into 96-well plates at a density of 1 × 10^4^ cells/well. After 16–24 h, cells were treated with the extracts from petioles and roots of *R. rhaponticum* and *R. rhabarbarum* and stilbenes (RHPG and RHPT), at concentrations of 1–100 µg/mL, for 24 h. After incubation, the cell culture medium was removed, and wells were rinsed twice with 0.02 M phosphate-buffered saline (PBS) containing Ca^2+^/Mg^2+^ (0.8 mM/0.4 mM), incubated in PBS containing Ca^2+^/Mg^2+^, 5.5 mM glucose, and 0.0125 mg/mL resazurin. HUVECs viability was estimated by measurements of the ability of live cells to reduce non-fluorescent resazurin to resorufin, a fluorescent product. After a 3 h incubation, resorufin fluorescence was measured (λ_ex_ = 530 nm, λ_em_ = 590 nm), using the Fluoroscan Ascent microplate reader (Thermo Fisher Scientific, Waltham, MA, USA) [20]. The metabolic activity of control HUVECs (untreated with the examined extracts and stilbenes) was assumed as 100% of cell viability. Samples treated with 1% Triton-X100 were reference samples, with no viable cells (0% of viability). In cell samples treated with the examined extracts or stilbenes, a decrease in cell viability ≥ 20% (compared to control/untreated HUVECs) was assumed as a cytotoxic effect.

### 2.5. Evaluation of the Anti-Inflammatory Properties of the Rhubarb-Derived Substances

#### 2.5.1. Analysis of the Inflammatory-Activated HUVEC Cytokine Profile

Cytokine and chemokine release from LPS-activated endothelial cells was analyzed using the Proteome Profiler Human Cytokine Array (Panel A) (R&D Systems, Minneapolis, USA), simultaneously detecting multiple analytes in cell culture supernatants. The applied panel allowed the identification of 13 substances, i.e., MIF (macrophage migration inhibitory factor), interleukin (IL)-8, Serpin E1, GM-CSF (granulocyte-macrophage colony-stimulating factor), GROα (growth-regulated oncogene α), IL-1α, IL-1β, IL-1ra, IL-6, MCP-1 (monocyte chemoattractant protein-1), MIP-1α (macrophage inflammatory protein 1α), RANTES (C-C motif chemokine ligand 5/regulated on activation, normal T cell expressed and secreted), and TNF-α (tumor necrosis factor alpha). The assay was performed using a cell culture medium, derived from HUVECs (1 × 10^6^ cells on a 6 cm dish), treated for 16 h with the tested substances (extracts or stilbenes), at the selected concentration of 5 µg/mL, followed by 4 h stimulation of the cells with LPS (1 µg/mL). Cell culture supernatants were collected after incubation, centrifuged, and mixed with the biotinylated detection antibody cocktail provided by the manufacturer. The samples were then incubated overnight with the membrane of the cytokine assay kit. After washing off the unbound material, the streptavidin–horseradish peroxidase conjugate and a chemiluminescent cytokine quantification reagent were added. Measurements were performed using a reader Syngen Biotech Azure 300.

#### 2.5.2. Adhesion of Monocytes to the Activated HUVECs

HUVECs were seeded into 6-well plates, at a density of 2 × 10^6^ cells/well. After 16–24 h, when the cells reached 100% confluency and formed paving stones, the medium was changed to a fresh one, containing 1–50 μg/mL extracts from the petioles of *R. rhaponticum* or *R. rhabarbarum*; 1–50 μg/mL extract from the roots of *R. rhaponticum*; 1–30 μg/mL extract from the roots of *R. rhabarbarum*; 1–50 μg/mL RHPT or 1–25 μg/mL RHPG. After 16 h, LPS (at the final concentration of 1 µg/mL) was added, and the incubation was continued for the next 4 h. Analysis of the adhesion of monocytes to the activated endothelial was performed as described previously [21]. U-937 cells were stained with Hoechst 33342 (1 μM), in the dark, for 10 min, and an excess of the fluorescent probe was removed by a series of RPMI-1640 medium washes. Then, the incubation medium from HUVECs was removed, and stained U-937 cells were added to the fresh cell culture medium (1 × 10^6^ monocytes per well). Incubation of monocytes with endothelial cells was performed in the dark, at 37 °C, and lasted 1 h. After that time, the unbound monocytes were removed by aspiration. The monolayers of HUVECs were rinsed twice with 0.02 M PBS containing Ca^2+^/Mg^2+^ (0.8 mM/0.4 mM). Finally, complexes of cells on the plate were fixed in 2% glutaraldehyde. Then, the stained monocytes were counted using a fluorescence microscope Zeiss (Jena, Germany). 

### 2.6. Evaluation of the Effects of the Examined Substances on COX-2 and 5-LOX Gene Expression

#### 2.6.1. Total RNA Isolation and cDNA Synthesis

Total RNA was isolated using the InviTrap Spin Cell RNA Mini Kit (Stratec Molecular, Berlin, Germany), following the protocol attached to the reagent kit. The purity of the samples (mRNA) was estimated based on optical density (OD) measurements in the BioTek Eon™ microplate spectrophotometer. The OD 260/280 value was >1.8, confirming RNA purity, and the OD 260/230 was >1.5, confirming the absence of residual solvents in the purified RNA. 

cDNA synthesis was performed with the PrimeScript RT Master Mix (Perfect Real Time, Takara; Kusatsu, Japan) according to the manufacturer’s instructions. 

#### 2.6.2. Real-Time Quantitative PCR (RT-qPCR)

Quantitative real-time PCR was performed using the Eco Real-Time PCR System (Illumina; San Diego, CA, USA). The total reaction volume (10 μL) consisted of 0.2 nM of forward and reverse primers, 1 μL cDNA template, 5 μL Takara BioSYBR Green Master Mix, and 3.6 μL DNAase/RNAase free water. The amplification conditions were as follows: an initial step of 95 °C for 30 s, then 40 cycles of 95 °C for 5 s, and 62 °C for 30 s. The gene-specific primers that were used are presented in Table 1. HPRT1 was used as a reference for gene expression normalization, performed according to the 2^−ΔΔCt^ method [22]. 

### 2.7. COX and LOX Inhibitor Screening 

Analysis of COX and LOX activity was carried out following the protocols of Cayman Chemicals (Ann Arbor, MI, USA). The Cyclooxygenase Colorimetric Inhibitor Screening Assay Kit (Cat. No. 701050) enables the measurement of the peroxidase component of COXs. The peroxidase activity was assayed colorimetrically by monitoring the appearance of the oxidized form of the *N,N,N′,N′*-tetramethyl-*p*-phenylenediamine (TMPD) at 590 nm. 

Lipoxygenase Inhibitor Screening Assay Kit (Cat. No. 760700) detects and measures the hydroperoxides produced in the lipoxygenation reaction using a purified LOX enzyme. Absorbance was measured at a wavelength of 500 nm. As a LOX inhibitor, nordihydroguaiaretic acid (NDGA) was used (at the concentration of 100 µM).

### 2.8. In Silico Studies: Bioactivity, Drug-Likeness, and Molecular Docking

Calculations of bioactivity and drug-likeness of the main compounds detected in *R. rhaponticum* and *R. rhabarbarum* extracts were completed by Molinspiration Cheminformatics website-calculation of Molecular Properties and Bioactivity Score-Predict Bioactivity tool (http://www.molinspiration.com/cgi–bin/properties accessed on 15 October 2021) and SwissTargetPrediction (http://www.swisstargetprediction.ch/ accessed on 15 October 2022). Structures of the compounds were used as ligands to predict binding to COX-2 and 5-LOX crystal structures in Autodock Vina 1.1.2 (http://vina.scripps.edu/ accessed on 12 May 2017) [23]. PDB coordinates of COX-2 (PDB ID: 4COX) [24] accessed on 26 December 2018, with bound indomethacin, and 5-LOX (PDB ID: 3V98, 3V99 and 6N2W) [25] were downloaded from the RCSB Protein Data Bank (http://www.rcsb.org/ accessed on 4 July 2021) [26]. The indomethacin-bound structure was implemented as a reference compound for COX-2 and docked to compute binding energy change for comparison with the experimental crystal structure complex. Structures of ligands were created from planar structures of known compounds drawn in ChemSketch Freeware 2018.2.1 (ACD/Labs: https://www.acdlabs.com/resources/free-chemistry-software-apps/chemsketch-freeware/ accessed on 24 April 2019), checked on the pages for the structures available on PubChem (https://pubchem.ncbi.nlm.nih.gov/ accessed on 4 July 2021), and ChemSpider (http://www.chemspider.com/ accessed on 4 July 2021), saved as the MOL format and translated to the MOL2 format using Open Babel 2.4.1 (http://openbabel.org accessed on 26 December 2018). Optimizations of the chemical structure’s geometries were calculated in Avogadro 1.2.0 (http://avogadro.cc accessed on 16 December 2018) [27] with the MMFF94 force field [28]. The ligands’ and proteins’ coordinates were prepared for docking in the ADT software from the MOL2 files as PDBQT files (http://autodock.scripps.edu/resources/adt accessed on 25 December 2018) [29]. The PDB files of all protein structures were purged manually of hetero atoms (HETATM). 10-fold dockings and subsequently parsing of energy results for all compounds were automated by scripts prepared in Python. Autodock Vina docking volume of COX-2 (4COX structure) covered boxes of two similar, opposite indomethacin binding sites with coordinates x, y, z in the centers: 24.864, 24.048, 10.330 and 69.785, 20.297, 7.825, respectively. Coordinates of cube centers for 5-LOX were set as −32.736, 78.963 and 8.039 according to the 3V98 structure and included inside the iron cation complexed with histidine residues as well as binding places of arachidonic acid and NDGA. The dimensions of the boxes were set as 26, 26, and 26 to embrace the active sites of both enzyme molecules. The interactions of ligands and amino acid residues within the binding pocket of the active site were analyzed by LigPlot+ v.2.2 (https://www.ebi.ac.uk/thornton-srv/software/LigPlus/ accessed on 22 December 2022) to generate ligand–protein interaction diagrams [30,31]. The visual assessment and image creation of ligands’ docked poses with proteins were set in UCSF Chimera 1.15 (http://www.cgl.ucsf.edu/chimera/ accessed on 22 September 2022) [32] and ChimeraX 1.5 (https://www.cgl.ucsf.edu/chimerax/ accessed on 28 November 2022) [33,34].

### 2.9. Statistical Analysis

The statistical analysis was performed using the STATISTICA 13.0 PL software (StatSoft Inc., Tulsa, OK, USA). The first step of statistical analysis was the elimination of the uncertain data by the Grubbs’ tests (GraphPad Prism 5.01, San Diego, CA, USA). Next, the differences between groups were assessed by the non-parametric Wilcoxon test (for unpaired data), and the Student’s *t*-test was used for data with normal distribution. A probability *p* < 0.05 was considered statistically significant. All the values in this work are expressed as mean ± standard deviation (SD).

## 3. Results

### 3.1. Phytochemical Profile of the Examined Rhubarb Extracts

Table 2, Appendix A show the detailed phytochemical profiles of the petioles and roots of *R. rhaponticum* and *R rhabarbarum*, including the semi-quantitative data on the contents of the principal metabolites.

Qualitatively, the chemical composition of the petiole butanol extracts (Table 2, Appendix A) of *R. rhabarbarum* and *R. rhaponticum* were very similar. Over 50% of observed metabolites were identical between the two extracts. Generally, they represented typical classes of compounds observed previously in rhubarbs [35,36,37,38]. The main anthraquinones were emodin (Table 2 and Appendix A, peak 151) and its anthrone derivative (peak 149), as well as numerous hexosides of both these compounds, mainly occurring as malonic or succinic acid esters (peaks 103, 106, 111, 116, 118, 120). Compared to derivatives of emodin, metabolites derived from physcion (rheochrysidin/methoxy-emodin) were relatively infrequent and restricted to acetyl-hexoside (peak 131), malonyl-hexoside, and hexoside of anthrone (peaks 130 and 139) as well as several mixed dianthrones with emodin. A relatively rare anthraquinone, a methoxylated derivative of nataloe-emodin, was observed in both species as malonyl-hexoside (peak 123). Of particular interest may be the tentatively identified dihexosides of emodin dianthrones, which represent structures analogous to sennosides from medicinal rhubarbs, but often exist as malonic acid esters (for example, peaks 109, 125, 127, 128, 129, 132, 133). They were observed as multiple isobaric peaks, presumably due to variable stereochemistry and various attached hexoses. Similar dianthrones were also formed between molecules of physcion and emodin.

Naphthalene derivatives were represented by torachrysone compounds (peaks 115, 121, 148). Compounds from the stilbene group were also detected: hexosides of resveratrol (resveratroloside and piceid, peaks 42 and 71), piceatannol (astringin, peak 39), and rhapontigenin (rhaponticin, peak 70). Flavonoid glycosides were represented by several compounds mainly derived from quercetin, apigenin, kaempferol, isorhamnetin, and pinocembrin. Apigenin derivatives were observed only as *C*-glycosides (peaks 45, 48, 56), and the remaining flavonoids were present as *O*-glycosides. Moreover, a whole group of typical polyphenolic compounds found in plants was also observed—phenolic acid hexosides, catechins, lignans, amino acids, and simple organic acids. The petiole extracts also contained small amounts of polar lipids and free triterpenes. A sulfonated catechin (peak 24) was observed in *R. rhaponticum* but was completely missing from *R. rhabarbarum*. Similarly absent were also flavonol derivatives of hexuronic acid (peaks 47 and 65).

The phytochemical composition of root extracts was less similar between the species, yet nearly 40% of the identified compounds were identical (Table 2, Appendix A and Figure 1). Compared with the petioles, in the root extracts (Table 2, Appendix A) the diversity of anthraquinone derivatives was smaller. The main compounds of this group present in the roots were chrysophanol malonyl hexosides (peaks 224 and 240). Chrysophanol hexoside esterified with gallic acid was also present (peak 225). Additionally, smaller signals from emodin hexosides, free emodin and its anthrone and dianthrone, and physcion hexoside and mixed emodin dianthrones with chrysophanol and physcion were observed. As in the petioles, the same torachrysone derivatives were present in the roots of both *Rheum* species.

In contrast, the observed variety of stilbene derivatives in the roots was greater. As shown in Appendix A, Table 2, Appendix A this group appeared to be a dominant type of metabolite observed in the root extracts, with astringin, rhaponticin, and deoxyrhaponticin forming prominent peaks on the chromatogram. The leading derivatives of this group were hexosides of resveratrol (peaks 69 and 91), piceatannol (peak 84), rhapontigenin (peak 121), and also deoxyrhapontigenin (peak 191). These hexosides were accompanied by numerous other derivatives, including malonic, gallic, coumaric, and ferulic acid esters. Pentosides and di-hexosides of stilbenes were also detected. As expected, the pool of flavonoid glycosides in the root was limited to a few derivatives of quercetin (peaks 126 and 128), apigenin (peak 130), and naringenin (peak 106). Phenolic acid derivatives were mainly detected as gallic and hydroxybenzoic acid esters of their hexosides.

Principal metabolites mentioned above are summarized in Table 2, while extensive tables containing complete data can be found in the Appendix A.

A summary of qualitative similarities and dissimilarities between the extracts from both species and plant parts is shown in Venn diagrams in Figure 1 and Appendix A. 

The quantitative analysis of the prepared extract was complicated due to the lack of appropriate calibration reference substances. In the petiole extracts, for 179 identified compounds, 10 reference substances were commercially available, whereas, in the root/rhizome extracts with 346 identified metabolites, just 14 reference substances were available. For this reason, we decided to use a semi-quantitative approach employing a universal charged aerosol detector (CAD). However, the response of all aerosol evaporative detectors varies as a function of mobile phase composition [15]. During isocratic separation, all the analytes at the same concentration should produce identical responses. In contrast, the higher percentage of the organic solvent, the higher the signal for the analytes will be observed during gradient elution. In the case of rhubarb extracts, utilization of isocratic elution is not feasible because samples are very complex. There are two ways of correcting signal increases during the gradient elution. One possibility is to provide the detector with a constant mobile phase concentration throughout the analysis, using the secondary pump running a reverse gradient through a separate, identical column. Both columns’ outflows are mixed before the detector, providing a constant concentration of the mobile phase [39,40]. While relatively simple in application and providing nearly perfect results [39], this approach requires additional, carefully set equipment and uses significant volumes of solvents. As an alternative, the so-called 3D calibration can compensate for signal changes during the elution [15]. We used a modified version of the second approach, applying a linear correlation between the increase in the organic component of the mobile phase and the increase in calibration slope coefficient for 32 reference standards eluting at different times throughout the separation. This approach allowed for a relatively reliable semi-quantitation of the metabolites detected in the extracts. The test set of 10 metabolites analyzed at two concentration levels (4 and 40 µg/mL) indicated deviations not exceeding 30% in the worst cases (for chrysophanol-8-glucoside) but usually reaching 10–15%, which was acceptable for our purposes. Because the CAD signal is one-dimensional, semi-quantitative contents data were calculated only for well-separated chromatographic peaks containing one primary component. This condition was difficult to achieve due to the richness of metabolites in the extracts, mainly in the roots. In a few cases of multi-component peaks considered particularly important for the study, the CAD peak area was divided based on ion intensities in corresponding HR-MS peaks. However, the results obtained in this way had decreased accuracy.

### 3.2. Effects of the Examined Substances on HUVEC Viability 

The viability of HUVECs treated with extracts from the petioles and roots of *R. rhaponticum* and *R. rhabarbarum*, two stilbenes (RHPG and RHPT), and reference compounds (indomethacin and zileuton) at concentrations of 1–100 µg/mL was analyzed by the resazurin reduction assay (Figure 2). A 24 h treatment with extracts from the petioles from *R. rhabarbarum* and *R. rhaponticum* did not affect the viability of HUVECs in the concentration range of 1–100 µg/mL (Figure 2A). In contrast, in samples treated with extracts from the roots at concentrations higher than 50 µg/mL and 30 µg/mL for *R. rhaponticum* and *R. rhabarbarum,* respectively, a decrease in cell viability was observed (Figure 2B). A 24 h treatment with stilbenes up to 100 µg/mL for RHPT did not affect cell viability; however, RHPG treatment decreased cell viability at concentrations higher than 25 µg/mL (IC_50_ = 46.9 µg/mL) (Figure 2C). An incubation of HUVECs with the non-steroidal anti-inflammatory drug, indomethacin, at concentrations above 80 μg/mL (IC_50_ = 93.13 µg/mL), resulted in a sharp decrease in cell viability, while zileuton showed no effect in the range of the tested concentrations (Figure 2D). A decrease in cell viability ≥ 20% (compared to the control/untreated HUVECs) was assumed as a cytotoxic effect.

Additionally, microscopic analyses of the HUVECs treated with *R. rhabarbarum* root extracts (30 µg/mL) and RHPG (25 µg/mL) were performed. Images were taken under transmitted light (Figure 2E, panels: a, b, and c) as well as a fluorescence microscope (Figure 2E, panels a’–c’), using the Calcein-AM fluorescent probe (3 µM). The obtained results confirmed the cellular safety of the examined substances at the selected concentrations.

The viability test results verified the cellular safety of the examined substances. Based on above data, concentrations not affecting cell viability were chosen for further experiments (Table 3).

### 3.3. Evaluation of the Anti-Inflammatory Properties of the Examined Substances

Anti-inflammatory effects of the examined extracts and stilbenes were monitored at a cytophysiological level (cytokine release from HUVECs and interactions with monocytes) as well as at a molecular level of intracellular processes (the expression of pro-inflammatory enzyme genes).

To evaluate the effects of the examined substances on cytokine secretion and interactions with monocytes, HUVECs were pre-incubated with the extracts or stilbenes, and then stimulated with LPS. In these assays, the anti-inflammatory efficiency of the examined extracts and stilbenes was estimated by comparing the cytokine level or monocyte influx in the HUVEC samples pre-incubated with the plant substances and activated by LPS, to cells stimulated with LPS in the absence of the examined extracts and stilbenes. 

Analyses of COX-2 and 5-LOX gene expression required the use of two experimental models. The plant-derived substances are exogenous factors that may themselves influence gene expression, as a part of the cell adaptive response. Therefore, in the first of the used experimental models, the expression of COX-2 and 5-LOX was studied in HUVECs pre-incubated with the examined plant substances, without subsequent stimulation with LPS. This assay enabled verification if the examined extracts influenced the HUVECs at their physiological state (under physiological conditions, with no exposure to pro-inflammatory stimuli). Effects of the rhubarb extracts and stilbenes on COX-2 and 5-LOX gene expression were evaluated by comparison with control samples, i.e., native HUVECs (untreated with the plant substances or LPS).

The second experimental model was designed to study the anti-inflammatory properties of rhubarb extracts and stilbenes under inflammatory conditions (i.e., in the LPS-stimulated cells). HUVECS were pre-incubated with the extracts or stilbenes, and then stimulated with LPS. The anti-inflammatory action of the examined substances was evaluated by comparison of the COX-2 and 5-LOX genes expression in these samples to the gene expression in HUVECs treated with LPS in the absence of the rhubarb extracts or stilbenes.

#### 3.3.1. Effects of the Rhubarb-Derived Compounds and Stilbenes on the Cytokine Secretory Profile of Endothelial Cells

The cytokine secretory profile of HUVECs was analyzed using the Proteome Profiler Human Cytokine Array, allowing the detection of growth factors, chemokines, interleukins as well as other factors that are essential both for the development and modulation of the inflammatory response of different cells. HUVECs were treated for 16 h with rhubarb-derived extracts and stilbenes at the selected concentration of 5 µg/mL and then stimulated with LPS (1 µg/mL) to induce an inflammatory response. The obtained results indicated that the examined substances might modulate the pro-inflammatory response of the HUVECs. Visualization of the proteome profiler membrane (Figure 3A) revealed changes in the secretion of 9 (out of 13 disclosed) cytokines. RHPT, the petiole extracts of both rhubarb species and the root extract from *R. rhaponticum* completely inhibited the release of the following cytokines: CCL5/RANTES, CXCL10/IP-10, CXCL12/SDF-1, and IL-18/IL-IF4. Furthermore, about 80% inhibition of the CCL5/RANTES release was observed in HUVECs treated with RHPG (Figure 3C). However, the petiole extracts of both species increased IL-8 secretion by about 50% (Figure 3D). An enhanced release of IL-8 was also found in cells treated with the *R. rhaponticum* root extract.

In cells treated with RHPT, the release of G-CSF and CM-CSF was reduced by about 60% when compared to cells treated with LPS in the absence of the examined plant-derived substances. However, in those samples, an increase in the level of IL-8 and MIF by over 30% was observed as well (Figure 3C). A similar effect was also found in cells treated with *R. rhaponticum* and *R. rhabarbarum* extracts from the petioles, including a significant decrease in G-CSF and CM-CSF, increased IL-8 release and the detection of MIF in the cell supernatant. In addition to typical cytokines, the used assay enabled the detection of the Serpin E1/PAI-1 (plasminogen activator inhibitor-1), an important regulator of hemostasis, the complement pathway and extracellular matrix remodeling. In this case, the most active one was the *R. rhabarbarum* root extract, reducing this protein release by about 80% (Figure 3E).

#### 3.3.2. Effects of the Rhubarb-Derived Extracts and Stilbenes on Endothelial Cell–Monocyte Interactions

Interactions of endothelial cells and leukocytes play a crucial role in the development of inflammation and in the amplification of inflammatory processes. Therefore, the assessment of the anti-inflammatory activity of the examined rhubarb extracts and stilbenes also included studies on their effects on the recruitment of monocytes to the activated HUVECs. A 16 h pre-treatment of HUVECs with the rhubarb-derived extracts and stilbenes, followed by a 3 h incubation with (1 µg/mL) LPS, significantly decreased the recruitment of U-937 monocytes to the activated endothelial cells (Figure 4), as confirmed by the microscope observations of Hoechst 33342-stained monocytes (Figure 4A). The root extracts of both *R. rhaponticum* and *R. rhabarbarum* and the stilbenes (RHPG and RHPT) displayed similar effects (Figure 4B,C,F,G, respectively) on monocyte recruitment. However, no significant changes in endothelial cell–monocyte interactions were found in samples treated with the *R. rhaponticum* extract from the petioles at a concentration of 1 µg/mL (Figure 4D). On the other hand, in HUVECs treated with the *R. rhabarbarum* extract from the petioles, applied at the same concentration (i.e., 1 µg/mL), the inhibition of monocyte influx was observed (Figure 4E).

#### 3.3.3. Effects of the Rhubarb-Derived Extracts and Stilbenes on Cyclooxygenase (COX-2) Expression in HUVECs

Comparing the results in both experimental models, it was noted that the extracts from the petioles of *R. rhaponticum* and *R. rhabarbarum* affected COX-2 gene expression (Figure 5A,B). In the unstimulated HUVECs, the maximal observed decrease in mRNA level expression was about 30% (*p* < 0.05, Figure 5A). Some fluctuations in COX-2 gene expression were observed in cells pre-incubated with the *R. rhaponticum* and *R. rhabarbarum* root extracts. RHPT had no effect on COX-2 mRNA levels in the unstimulated cells (*p* > 0.05), and the RHPG suppressed COX-2 gene expression maximally by approximately 20-25% (*p* < 0.05, Figure 5A). 

In the LPS-activated cells (Figure 5B), a decrease in COX-2 mRNA levels in the presence of the petiole extracts and stilbenes was more evident. The gene suppression level in most samples of HUVECs treated with these preparations attained or even exceeded 50%. In contrast, the *R. rhaponticum* and *R. rhabarbarum* root extracts increased the level of COX-2 gene expression (Figure 5B). 

#### 3.3.4. Effects of the Rhubarb-Derived Extracts and Stilbenes on 5-Lipoxygenase (*ALOX5*) Gene Expression in HUVECs

In the unstimulated cells (Figure 6A), only the *R. rhaponticum* petiole extract influenced the *ALOX5* mRNA level in the full range of tested concentrations (1–50 µg/mL; * *p* < 0.01 and *** *p* < 0.001). The remaining substances displayed no effects or only slightly reduced the mRNA levels.

In the activated HUVECs (Figure 6B), extracts from the petioles of both rhubarb species and stilbenes markedly inhibited *ALOX5* gene expression, compared to this gene’s expression level in cells activated by LPS in the absence of the examined rhubarb-derived substances. The *R. rhaponticum* root extract decreased *ALOX5* gene expression only at its highest concentration, i.e., 50 µg/mL, whereas the *R. rhabarbarum* root extract did not affect this gene’s expression (Figure 6B).

### 3.4. COX-2 and 5-LOX Inhibitor Screening

To assess whether the examined extracts and selected components can acts as anti-inflammatory agents at a level of COX-2 and 5-LOX enzyme activity, a colorimetric inhibitor screening test was performed. The reference compound was indomethacin. 

The experiments revealed that the activity of the pro-inflammatory enzyme COX-2 was most effectively reduced by the root extracts of both species of rhubarb, used at a concentration of 50 μg/mL (Figure 7A, *** *p* < 0.001). In these samples, the enzyme activity was reduced by about 80%, when compared to the native (untreated) COX-2. The IC50 values for the extracts from the roots of *R. rhaponticum* and *R. rhabarbarum* were 19.16 μg/mL and 19.44 μg/mL, respectively. The COX-2-inhibitory effect (about 30% of enzyme activity reduction) was also found in the samples treated with RHPG at the same concentration (i.e., 50 μg/mL; *** *p* < 0.001). RHPT and extracts from the petioles of *R. rhaponticum* and *R. rhabarbarum* were considerably weaker inhibitors of this enzyme (Figure 7A).

In the case of 5-LOX, most of the examined substances displayed slight inhibitory activities (Figure 7B). The most effective was RHPG, at a concentration of 1 µg/mL (*** *p* < 0.001). The reference inhibitor was nordihydroguaiaretic acid (NDGA), contained in the kit at a concentration of 3 µg/mL.

### 3.5. In Silico Studies: Bioactivity, Drug-Likeness, and Molecular Docking 

The most common organic chemical compounds identified in the plant extracts were subjected to computational drug-likeness analysis and their ability to bind to COX-2 and 5-LOX protein molecules in the vicinity of their active sites as potential competitive inhibitors (Table 4). All compounds generally met the criteria for drug candidates by the Molinspiration Molecular Properties and Bioactivity Score. According to SwissTargetPrediction some of them were probable targets for COX-2 or 5-LOX (Appendix A in the Appendix A). All of them showed a significant ability to bind to COX-2 but only a few had an affinity for 5-LOX in Autodock Vina molecular docking. Many of them, including piceatannol-galloylglucoside, viniferin (resveratrol-dehydrodimer), emodin, emodin anthrone, deoxyrhaponticin, rhaponticin, resveratrol 3-*O*-beta-glucopyranoside, digalloyl glucoside, piceid (polydatin), chrysophanol-8-glucoside, and astringin, showed particularly high negative values for the change in binding energy in the active site (range from −9.3 to −8.5 kcal·mol^−1^), indicating an exoergic process of high-affinity, compared to indomethacin (−9.4 kcal·mol^−1^), an inhibitor and non-steroidal anti-inflammatory drug. In contrast, most compounds bound weakly at the 5-LOX active site, often with positive changes in binding energy (Table 4).

Due to the high content of rhaponticin in the examined extracts (Table 2), worthy docking results, and drug-likeness parameters, we focused further on the analysis of computational results obtained for this compound compared to known inhibitors, such as indomethacin and NDGA, and the natural substrate arachidonate. Rhaponticin fits in the COX-2 hydrophobic binding pocket of the substrate arachidonate, in the area where also non-steroidal anti-inflammatory drugs and inhibitors, such as indomethacin and NDGA, attach. According to LigPlot+ v.2.2 and visual inspection on UCSF Chimera and ChimeraX rhaponticin conformers bound near amino acid residues Tyr385, Ser530, Glu524, and Arg120 stabilized by polar interactions, such as hydrogen bonds, van der Waals interactions, and hydrophobic effects, with Val116, Val359, Leu531, and other residues (Figure 8).

According to the 4COX crystallographic structure, bound indomethacin in the chain A binding pocket site 1 could interact by hydrogen bonding with the Arg120 side chain -NH1 (hydrogen donor), Tyr355 (hydroxyl group donor) - ligand carbonyl oxygen of carboxyl group as an acceptor. The Ser353 hydroxyl group could be the acceptor and donor with the carboxyl of the ligand. The Ser 530 hydroxyl of the side chain forms a hydrogen bond with the carbonyl oxygen of indomethacin, and methoxy oxygen can be an acceptor of the hydroxyl hydrogen of Tyr355. Many van der Waals and hydrophobic effects could be important for this ligand stabilization including: the indol group with Val523, Ser353, Ala527, Phe518, Val349, Leu352, a phenyl ring with Phe381, Trp387, Tyr385, Gly526, Leu384, Met522, Ser530, and the methyl group Leu531. The Gly526 residue could interact with an indomethacin chlorine atom (Figure 8A). Indomethacin and NDGA docked in the same location showing similar interactions (Figure 8B,C) with rhaponticin.

## 4. Discussion

From ancient times to modern history, plants (including different rhubarb species) have been used to treat or support the treatment of different diseases. *R. rhabarbarum*-based preparations were administered as purgative agents as well as to treat liver, spleen, and stomach dysfunctions. Traditional medicine recommendations for the therapeutic use of *R. rhaponticum* include purgative effects, gastrointestinal and reproductive system disorders, injuries, heartache, and pain in the pericardium [7]. Currently, *R. rhaponticum* is primarily known as a component of the preparations dedicated to alleviating menopausal complaints, especially the ERr731^®^ extract, which has been registered as a medicinal product. The phytoestrogenic activity of the preparation based on the rhapontic rhubarb rhizome has become the leading research trend on this species. In contrast, other biological activities of both the aforementioned species have been elucidated and described to a significantly lesser extent. A rich phytochemical profile of rhubarbs containing different groups of flavonoids, stilbenes, phenolic acids, anthraquinones, and naphthalene derivatives suggests the possibility of beneficial, pleiotropic activity through different molecular pathways in the human body [4].

The selection of raw materials with diversified and the highest possible content of bioactive ingredients is very important in both functional food production and for medicinal purposes as well. Most studies on the chemical composition of *Rheum rhabarbarum* or *Rheum rhaponticum* mainly focus on the petioles/stalks, consumed as a vegetable and used in culinary preparations, such as soups, pies, tarts, jams, jellies, compotes, juices, candies, and others [41]. Therefore, one of the goals of this study was to screen, fully characterize the possible components in complex samples and quantify the principal compounds in *Rheum rhabarbarum* and *Rheum rhaponticum* organs, such as the petioles and roots, which will enable chemical fingerprinting and metabolomic identification of these species. According to previous studies [35,36,37,38], we also identified several metabolites from the most important biologically active classes, including anthraquinones, anthrones, and phenolic compounds: stilbenes, flavonoids, phenolic acids, cinnamic acid derivatives, and tannins. Despite superficial similarities in qualitative comparisons, the two investigated species appear to differ significantly in their contents of active ingredients. The contents of rhaponticin in the petioles and deoxyrhaponticin in the roots best illustrate this observation. In the petioles, garden rhubarb contains 10 times more rhaponticin than rhaponic rhubarb (Table 2). Furthermore, the petioles of garden rhubarb also contain an isomer of rhaponticin, only present in rhapontic rhubarb in minuscule quantities. Similarly, the levels of deoxyrhaponticin in the roots of *R. rhabarbarum* are much higher than in the roots of *R. rhaponticum*. The levels of rhaponticin are similar in these organs and do not seem to correlate with the levels of deoxyrhaponticin. Therefore, deoxyrhaponticin is presumably not a derivative of rhaponticin but an independently synthesized metabolite.

The present work evaluated the anti-inflammatory effects of plant extracts from two popular, edible rhubarbs (*R. rhaponticum* and *R. rhabarbarum*) in the context of their cardioprotective potential. The anti-inflammatory properties of rhubarb extracts isolated from the petioles and underground organs of *R. rhabarbarum* and *R. rhaponticum*, as well as of the two stilbenoids usually found in these plants, i.e., rhapontigenin and its glycoside, rhaponticin, were examined using an experimental model of vein endothelial cells (HUVECs). The endothelial cells are not only an integral part of blood vessel physiology and vascular homeostasis, but also participate in many other physiological and pathological processes, including cardiovascular diseases. Numerous studies have suggested that dietary plant-derived substances, especially those with antioxidant activity, may be a helpful and promising strategy for preventing cardiovascular disorders [42,43]. On the other hand, a growing number of critical opinions have emphasized that the antioxidant action is one of many mechanisms of beneficial effects of plant-derived compounds on the human body. Inflammation, endothelial dysfunction, and changes in blood physiology form a closely related network of pathological interactions resulting in pro-inflammatory and adhesive properties of the vessel wall. Inflammation is a key event in the pathophysiology of various cardiovascular disorders - it affects the functionality of both the blood vessel wall and blood components. For this reason, besides the antioxidant action, other properties of natural substances, including their anti-inflammatory actions may be pivotal for their health-promoting or even therapeutic effects [44,45,46].

The first step of the study involved cytotoxicity tests and confirmed cellular safety of the *R. rhaponticum* extracts from the roots and petioles, *R. rhabarbarum* petiole extract, and rhaponticin, at concentrations of 1–50 µg/mL. In the case of *R. rhabarbarum* root extracts and rhapontigenin, a safe concentration range was lower and amounted to 1–30 and 1–25 µg/mL, respectively. Application of the examined extracts at the mentioned concentrations corresponds to nano- or micromolar concentrations of their bioactive ingredients in the tested samples, which is consistent with the literature data on physiologically relevant levels of the plant-derived substances. It has been established that the blood plasma levels of natural plant-derived compounds or their metabolites achieve concentrations from nanomoles to a few micromoles per liter [47].

The literature provides very little information on the anti-inflammatory actions of extracts derived from garden and/or rhapontic rhubarb. An aqueous extract from *R. rhabarbarum* suppressed the tumor necrosis factor α-induced activation of NF-κB-p65 as well as the expression of adhesion molecules (ICAM-1 and VCAM-1) and the monocyte chemoattractant protein-1 (MCP-1) [48]. More data has been reported from studies focused on rhubarb-derived stilbenes or other single compounds. For instance, six stilbenes (rhaponticin, rhapontigenin, isorhaponticin, desoxyrhaponticin, desoxyrhapontigenin and resveratrol) isolated from the *R. rhabarbarum* rhizome reduced the reactive oxygen species production in RAW 264.7 macrophages [49]. The ability of rhaponticin (1 µM) to inhibit the MAPK/NF-κB signaling pathways in LPS-stimulated endothelial cells has been recently reported [50]. It has been suggested that the anti-inflammatory action of natural stilbenoids (pinosylvin, monomethylpinosylvin, resveratrol, pterostilbene, piceatannol, and rhapontigenin) is possibly mediated via inhibition of the PI3K/Akt pathway [51]. The PI3K/Akt pathway is activated in low-grade inflammation [52], commonly associated with chronic inflammatory processes occurring in cardiovascular disorders, e.g., atherosclerosis. Therefore, an effective modulation of this pathway by natural substances provides a promising basis for further research. In other work, aloe-emodin from *R. rhabarbarum* suppressed LPS-induced iNOS expression, degradation of IκBα and phosphorylation of ERK, p38, JNK, and Akt in macrophages [53].

Our experiments indicated that the examined extracts and substances possessed anti-inflammatory properties and modulated the pro-inflammatory response of endothelial cells at the level of gene expression, cytokine release and interactions with leukocytes. However, the activities of the preparations from the petioles and root extracts differed significantly. For example, extracts from the petioles of *R. rhaponticum* and *R. rhabarbarum*, as well as RHPG and RHPG significantly reduced the expression of the *COX-2* gene in LPS-activated HUVECs. On the other hand, the roots extracts increased the level of gene expression. Similar diversity in anti-inflammatory efficiency was observed in 5-LOX (*ALOX5*) gene expression. The *R. rhaponticum* and *R. rhabarbarum* petiole extracts, together with RHPG and RHPG, evidently reduced the expression of the *ALOX5* gene, while the root extract from *R. rhaponticum* suppressed the *ALOX5 gene* only at the highest used concentration (i.e., 50 μg/mL), and the *R. rhabarbarum* root extract was inefficient at this level of modulation of the 5-LOX enzyme action.

The inhibitor screening tests, carried out in the next stages of the present study, demonstrated that the examined substances might also act at the level of the active enzyme COX-2 and partly reduced its activity. The examined substances inhibited COX-2 enzyme activity with varied efficiency. Their inhibitory activity towards 5-LOX was significantly weaker. This observation was consistent with results obtained in silico, where most of the tested compounds showed much better COX-2 inhibitory abilities than 5-LOX, referring to such chemical compounds as piceatannol-galloylglucoside, viniferin, emodin, emodin anthrone, deoxyrhaponticin, rhaponticin, resveratrol 3-*O*-beta-glucopyranoside, digalloyl glucoside, piceid and, astringin, and chrysophanol-8-glucoside. Rhaponticin was most abundant in the roots of both rhubarb species (184.0 ± 10.93 and 151.3 ± 7.77 mg per g of extract, respectively), one to two orders of magnitude more than the other compounds (Table 2). Taking into account that the root extracts of both species of rhubarb clearly inhibited COX-2, especially visible at the highest concentration used, and the docking indicated strong binding to the active site (−8.5 ± 0.2 kcal·mol^−1^), with no such effect of 5-LOX, it might be assumed that it was the most important compound responsible for the selective anti-inflammatory effect (Figure 8) binding in COX-2 hydrophobic pockets where arachidonate attaches as a substrate.

The COX-2 hydrophobic pocket is important for arachidonate hydrocarbon chain binding with amino acid residues: Val116, Arg120, Leu352, Gly326, Leu 359, Leu 531, Leu93, Val523, and Tyr385 (Figure 8D). Tyr385 is a key player in catalytic activity. A hydroperoxide oxidizes the heme to a ferryl-oxo derivative and Tyr385 forms a tyrosyl radical which attacks arachidonate, oxidizing and detaching its 13-pro(S) hydrogen to initiate the COX cycle. Blocking of the Tyr385 residue by non-steroidal anti-inflammatory drugs inhibits the whole arachidonate transformation process. Rhaponticin and other stilbene-carbohydrate derivatives, located in the hydrophobic pocket and stabilized by hydrogen bonding, might act as inhibitors analogous to non-steroidal anti-inflammatory drugs, such as indomethacin and NDGA, occupying this place and blocking it from binding and oxidizing arachidonate (Figure 8).

Conversely to the results obtained at the level of pro-inflammatory enzymes gene expression, at the level of enzyme activity, the root extracts were more effective than extracts isolated from the petioles. The IC_50_ values for extracts from the roots of *R. rhaponticum* and *R. rhabarbarum* were 19.16 µg/mL and 19.44 µg/mL, respectively. The IC_50_ values for RHPG and RHPT attained concentrations > 50 µg/mL. Our work provides the first evidence of the ability of rhubarb extracts to inhibit the COX-2 enzyme. However, some information on the cyclooxygenase inhibitory effects is available for RHPG. Based on the in vitro evaluation of the effect of dietary stilbenes on 5-lipoxygenase and cyclooxygenase activities [54], Kutil and co-authors (2015) reported that this stilbene can inhibit COX-2 activity (IC_50_ = 36.1 µM). On the other hand, our in vitro studies on the stilbene effects on 5-LOX activity suggested weak inhibitory efficiency of this enzyme of both RHPG and RHPT (IC_50_ > 50 µg/mL). According to the literature, the LOX-inhibitory efficiency of rhapontigenin was characterized by an IC_50_ of 10.7 µM, and for rhaponticin, the IC_50_ was 34.3 µM [55,56].

Furthermore, our work demonstrated that the examined extracts and stilbenes might reduce or modulate the release of pro-inflammatory cytokines from HUVECs. Results of the HUVEC cytokine profiling demonstrated that most of the examined substances (including RHPT, the petiole extracts of *R. rhaponticum* and *R. rhabarbarum*, and the root extract from *R. rhaponticum*) inhibited the release of crucial pro-inflammatory factors such as CCL5/RANTES, CXCL10/IP-10, CXCL12/SDF-1, and IL-18/IL-IF4. On the other hand, an increase in the IL-8 level was found for both stilbenes and most of the examined extracts (except for the *R. rhabarbarum* root extract). This latter observation may be a result of both a divergence in mechanisms of action of the examined plant-derived substances and the complexity of the immune response, including multiple pathways of control for IL-8 synthesis and secretion [57]. Regulation of the synthesis and release of IL-8 by natural substances has only partly been elucidated, and the available evidence is inconsistent. For example, a reduction in IL-8 release from THP-1 monocytes was demonstrated for *Castanea sativa* Mill., *Alchemilla vulgaris* L., and *Salix alba* L. [58]. However, in another study, green tea extracts induced IL-8 mRNA and protein expression in Caco-2 cells [59]

Chronic and severe inflammatory processes occurring in the cardiovascular system influence the physiology of endothelial cells and impair their modulatory functions. The endothelium is not only a regulator of blood hemostasis, but also a modulator of vascular tone, redox balance, blood flow and fluidity, as well as immune response. For this reason, endothelial dysfunction is considered a systemic pathological state of the vasculature. The inflammatory-activated endothelial cells lose their anti-coagulant and modulatory properties in favor of pro-coagulant activity and pro-atherogenic effects. They also release numerous cytokines, chemokines, and growth factors, leading to an augmentation of the inflammatory response and an increase in endothelium permeability [11,60,61].

Our work provides new evidence of the biological activity of rhubarb-derived species and indicates that the examined plant extracts and stilbenes display anti-inflammatory effects. Using a combination of in vitro studies and in silico analyses, we have demonstrated, for the first time, the anti-inflammatory potential of rhubarb extracts and stilbenes in the context of endothelium physiology and the prevention of its dysfunction. The examined plant-derived substances displayed a protective action towards the undesired, inflammation-triggered changes in the endothelium physiology. They reduced the endothelial cell pro-inflammatory response at different molecular levels. Rhubarb-derived extracts-displayed the ability to modulate the pro-inflammatory gene expression, inhibited the COX-2 enzyme activity, suppressed the release of pro-inflammatory cytokines, and reduced the macrophage influx to endothelial cells. The obtained results provide a promising background for further studies, including in vivo work. 

## Figures and Tables

**Figure 1 nutrients-15-00949-f001:**
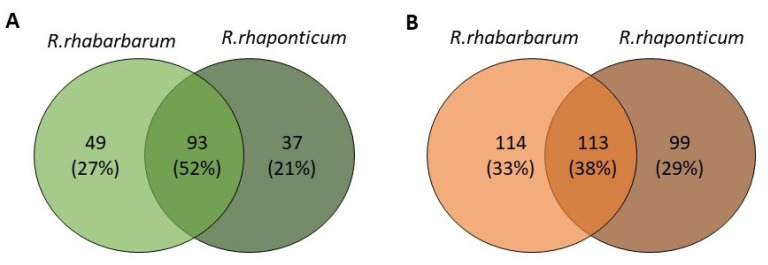
Qualitative similarities between the extracts from the petioles (**A**) and roots (**B**) of *R. rhabarbarum* and *R. rhaponticum*. Numbers indicate common and unique metabolites for each species.

**Figure 2 nutrients-15-00949-f002:**
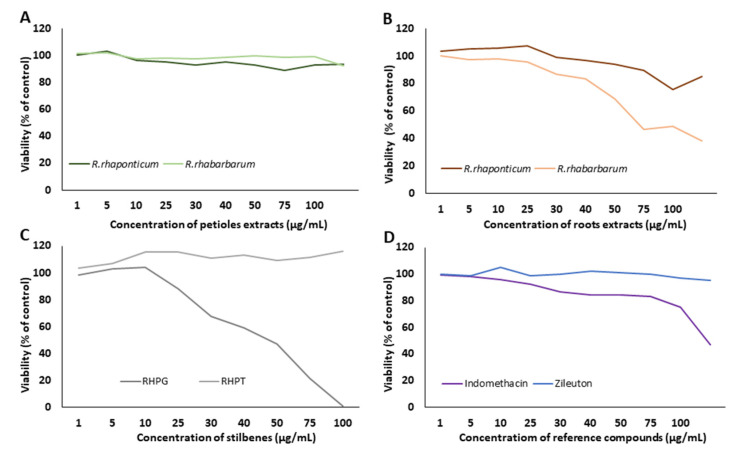
Effects of the examined substances on HUVEC viability. The figure presents the viability curves of HUVECs after 24 h of incubation with the petiole extracts from *R. rhaponticum* and *R. rhabarbarum* (**A**), the root extracts from *R. rhaponticum* and *R. rhabarbarum* (**B**), RHPG and RHPT (**C**) and reference anti-inflammatory drugs (indomethacin and zileuton) (**D**), at concentrations of 1–100 µg/mL, established in the resazurin-based assay. Results derive from four independent experiments. Panel (**E**) presents the effects of *R. rhabarbarum* root extract (**b**,**b′**) and RHPG (**c**,**c′**) at concentrations of 30 and 25 µg/mL, respectively. The (**a**,**b**,**c**) panels are the images of HUVECs in transmitted light, whereas the (**a′**,**b′**,**c′**) panels represents the same HUVECs samples obtained from the fluorescence microscope. HUVECs were stained with 3 µM Calcein-AM; images were acquired with 4× magnification.

**Figure 3 nutrients-15-00949-f003:**
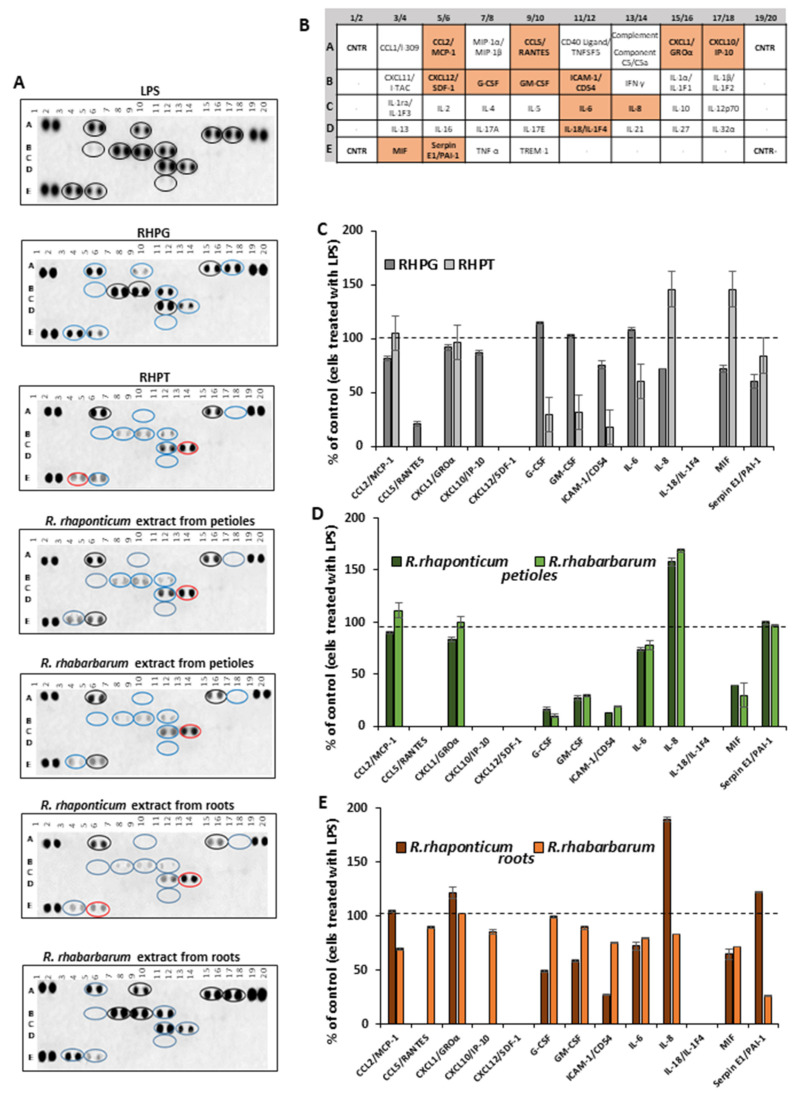
Effects of the examined rhubarb extracts and stilbenes on the HUVEC cytokine secretory profile, detected by the Proteome Profiler Human Cytokine Array. Panel (**A**) contains a representative dot-blot pattern of cytokines, released into the cell culture medium by HUVECs pre-incubated with the rhubarb extracts or the stilbenes (5 µg/mL), and stimulated by LPS (1 µg/mL). Panel (**B**) refers to the table for the Human Cytokine Array coordinates. The graphs show results calculated into percentage, based on the positive control (i.e., cells treated with LPS in the absence of the examined substances), which was assumed as 100% of cytokine secretion. Panel (**C**)—data for stilbenes; panel (**D**)—results for the petiole extracts; panel (**E**)—results for the root extracts.

**Figure 4 nutrients-15-00949-f004:**
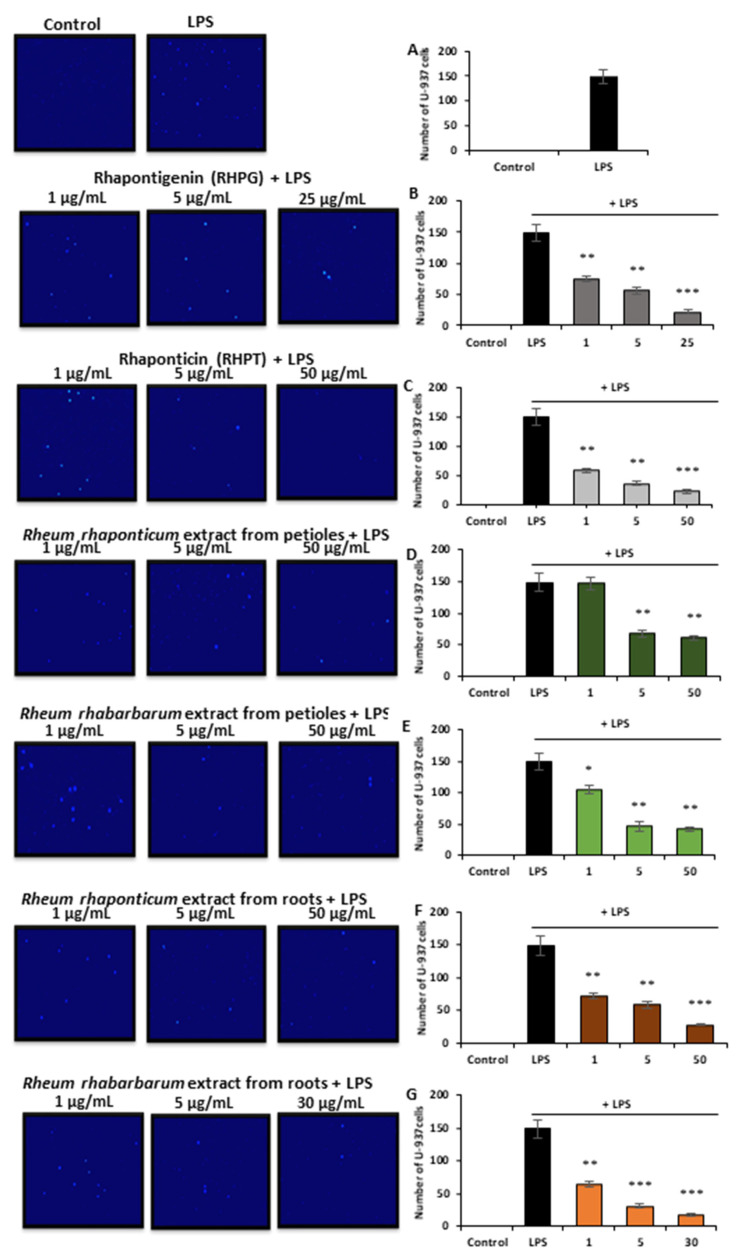
The effect of the examined rhubarb extracts and stilbenes on the recruitment of U-937 monocytes to the LPS-activated HUVECs. The image panel represents the pictures obtained from the fluorescence (UV) microscope; magnification 20×. Calculations of the adhered monocytes have been presented as follows: control/untreated and LPS-treated cells (**A**), RHPG (**B**), RHPT (**C**), *R. rhaponticum* extract from the petioles (**D**), *R. rhabarbarum* extract from the petioles (**E**), *R. rhaponticum* extract from the roots (**F**), *R. rhabarbarum* extract from the roots (**G**). Data are presented as means ± SD; *n* = 3. Statistical significance (HUVECs pre-incubated with plant substances and activated by LPS versus HUVECs activated by LPS in the absence of the examined substances): * *p* < 0.05, ** *p* < 0.01, *** *p* < 0.001.

**Figure 5 nutrients-15-00949-f005:**
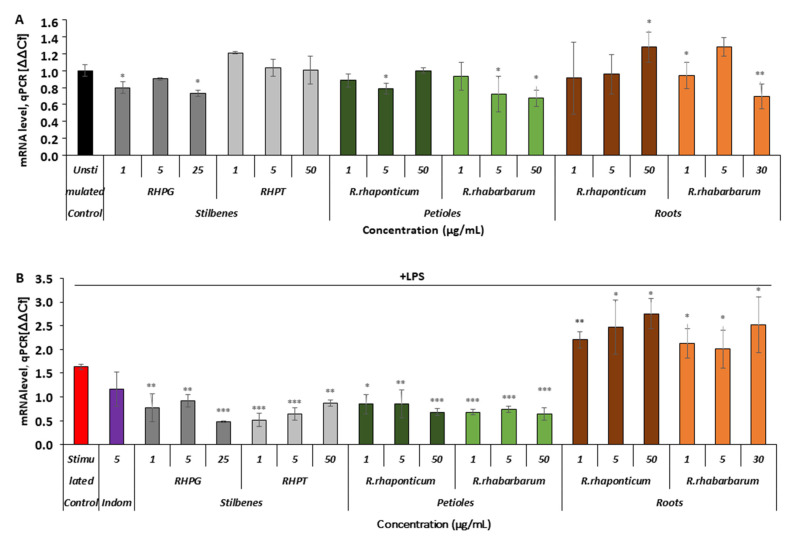
Effects of the rhubarb extracts and stilbenes on COX-2 gene expression in the unstimulated and inflammatory-activated HUVECs. Panel (**A**)—data derived from experiments involving HUVECs treated with stilbenes (RHPG and RHPT) or rhubarb extracts, without subsequent pro-inflammatory stimulation with LPS. Panel (**B**)—COX-2 gene expression in HUVECs treated with the stilbenes (RHPG and RHPT) or rhubarb extracts and stimulated with LPS; *n* = 5; * *p* < 0.05; ** *p* < 0.01; *** *p* < 0.001. Indom—indomethacin, a reference compound (COX inhibitor).

**Figure 6 nutrients-15-00949-f006:**
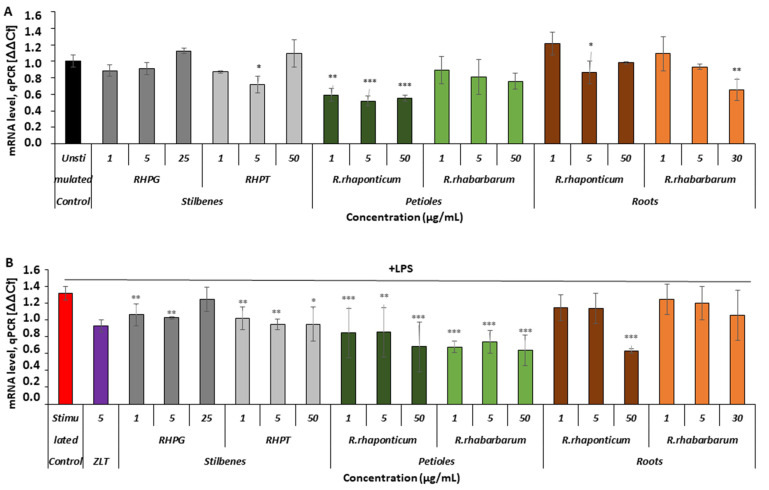
Effects of the rhubarb extracts and stilbenes on *ALOX5* gene expression in unstimulated and inflammatory-activated HUVECs. Panel (**A**)—*ALOX5* gene expression in HUVECs treated with stilbenes (RHPG and RHPT) or rhubarb extracts, without subsequent pro-inflammatory stimulation with LPS. Panel (**B**)—*ALOX5* gene expression in HUVECs pre-incubated with stilbenes (RHPG and RHPT) or rhubarb extracts, and stimulated with LPS; *n* = 5; (* *p* < 0.05; ** *p* < 0.01; *** *p* < 0.001). ZLT -zileuton, a reference compound (5-LOX inhibitor).

**Figure 7 nutrients-15-00949-f007:**
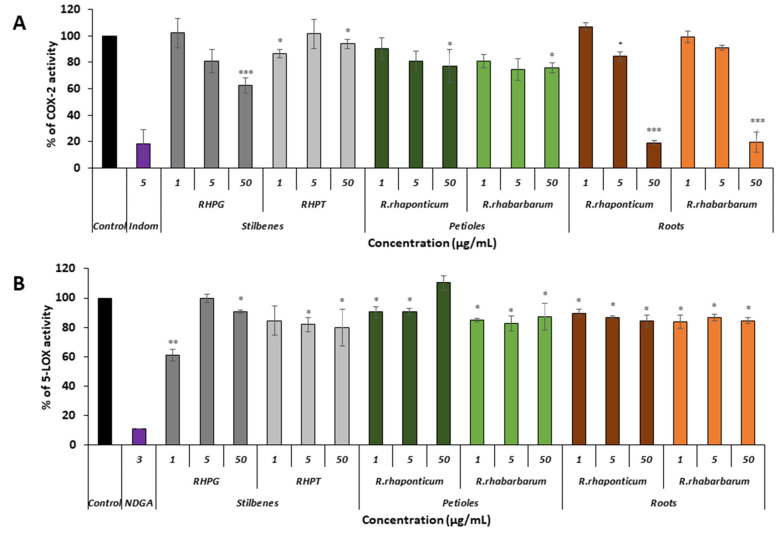
Effect of stilbenes (RHPG and RHPT) as well as the petiole and root extracts of *Rheum rhaponticum* and *Rheum rhabarbarum* on the activity of COX-2 (panel **A**) and 5-LOX (panel **B**); *n* = 7. The activity of native enzymes (untreated with any of the examined substances) was assumed as 100%; * *p* < 0.5; ** *p* < 0.01; *** *p* < 0.001.

**Figure 8 nutrients-15-00949-f008:**
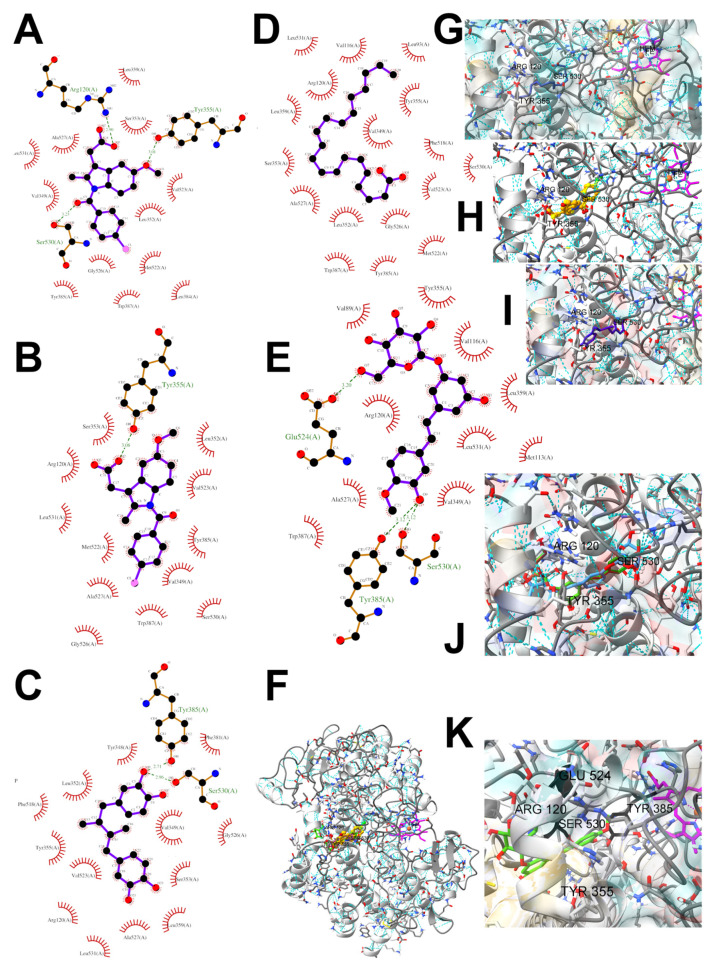
Ligand binding analysis in the area of hydrophobic COX-2 active site pocket 1 according to the crystal structure 4COX where the location of indomethacin was determined experimentally. The panels from A to F are LigPlot+ v.2.2 analysis: (**A**) experimental indomethacin binding with amino acid residues in the 4COX crystal structure, (**B**) docking studies of indomethacin, (**C**) NDGA docking, (**D**) arachidonate docking (**E**) rhaponticin docking. The panels from F to K show the spatial structure of the COX-2 monomer. (**F**) A view of the whole monomer-bound indomethacin molecules (yellow, orange, crystal structure, and docked, respectively), rhaponticin molecules (green and plum) and arachidonate (khaki). Heme moiety of the peroxidase site (pink) with an iron cation (orange) visible on the right. The panels from G to K show the zoomed-in substrate–inhibitor binding site with ligands (**G**) arachidonate (light navy) (**H**) indomethacin molecules (yellow, orange, crystal structure, and docked, respectively), (**I**) NDGA (dark navy) (**J**) rhaponticin 10 molecules (**K**) rhaponticin bound—one molecule (light green). Hydrogen bonds are marked by light blue dashed lines. Glu536, Ser530 and Tyr385 are marked in dark gray.

**Table 1 nutrients-15-00949-t001:** Primers sequence used in mRNA qPCR.

Gene	Forward Primer 5′ > 3′	Reverse Primer 5′ > 3′	Amplicon Size (bp)
*HPRT1*	ATGGACAGGACTGAACGTCTT	TCCAGCAGGTCAGCAAAGAA	113
*COX2*	GCACTGTTRGGTGGGT	AGAAAACTGCTCAACACCGGAA	94
*ALOX5*	CGATACTTATGAAAGGCCCAGACC	GGTCTGGGAGACCGTACTGGA	83

**Table 2 nutrients-15-00949-t002:** Main metabolites detected in the butanol fractions of the petioles and roots of *R. rhabarbarum* and *R. rhaponticum*. A complete dataset is available in Appendix A.

No	Name	RT (min)	Calc. Formula	Err. (ppm)	*R. rhabarbarum* Petioles (mg/g d.w.)	*R. rhaponticum* Petioles (mg/g d.w.)	ID Level
2	*Unidentified*	2.07	C_10_H_12_O_10_	0.7	5.06 ± 1.06	1.71 ± 0.27	4
3	*Unidentified*	2.07	C_10_H_12_O_11_	0.7	6.99 ± 1.47	1.97 ± 0.30	4
4	beta-glucogallin	2.22	C_13_H_16_O_10_	1.4	14.67 ± 3.79	0.85 ± 0.19	1
5	homocitrate-1	2.86	C_7_H_10_O_7_	2.8	2.49 ± 1.33	1.07 ± 0.29	3
6	homocitrate-2	3.09	C_7_H_10_O_7_	2.2	7.16 ± 2.01	5.40 ± 0.90	3
7	gentisoyl-Hex	3.70	C_13_H_16_O_9_	−0.2	ND	1.59 ± 0.12	3
11	Trp	6.46	C_11_H_12_N_2_O_2_	−0.9	2.63 ± 0.66	ND	2
13	syringoyl-Hex-2	7.57	C_15_H_20_O_10_	0.6	1.36 ± 0.39	4.21 ± 0.99	3
14	procyanidin-B1	7.76	C_30_H_26_O_12_	0.6	1.60 ± 0.44	0.05 ± 0.02	1
16	*Unidentified*	8.51	C_12_H_22_O_8_	−0.8	ND	1.33 ± 0.34	4
17	catechin	8.81	C_15_H_14_O_6_	0.7	24.71 ± 3.36	0.26 ± 0.05	1
23	eucomic-acid-2	9.56	C_11_H_12_O_6_	0.0	1.79 ± 0.35	0.40 ± 0.07	2
25	*Unidentified*	9.93	C_8_H_14_O_5_	−0.8	0.43 ± 0.03	1.43 ± 0.21	4
31	sinapoyl-Hex-2	11.31	C_17_H_22_O_10_	0.0	3.53 ± 0.77	ND	3
35	methyl-butyl-Hex-Pent	12.09	C_17_H_32_O_12_	−0.1	1.30 ± 0.23	ND	3
42	resveratrol-Hex-1 (resveratroloside)	13.24	C_20_H_22_O_8_	1.0	1.24 ± 0.17	ND	2
43	tetrahomocitrate	13.43	C_10_H_16_O_7_	1.6	0.33 ± 0.18	0.88 ± 0.10	3
44	*Unidentified*	13.69	C_21_H_24_O_12_	0.7	0.77 ± 0.12	ND	4
46	myrcetin-Hex-dHex	14.08	C_27_H_30_O_17_	1.7	3.15 ± 0.73	1.21 ± 0.08	3
47	myrcetin-HexA	14.10	C_21_H_18_O_14_	0.2	ND	0.74 ± 0.05	3
48	vicenin-III	14.14	C_26_H_28_O_14_	1.2	1.76 ± 0.41	1.74 ± 0.11	2
54	*Unidentified*	14.69	C_21_H_22_O_12_	1.0	0.82 ± 0.16	ND	4
55	galloyl-catechin-1	15.02	C_22_H_18_O_10_	0.9	0.63 ± 0.18	0.21 ± 0.01	2
62	rutin	15.83	C_27_H_30_O_16_	1.8	17.69 ± 1.85	11.60 ± 0.95	1
65	quercetin-HexA	16.02	C_21_H_18_O_13_	0.3	ND	4.66 ± 0.21	3
66	quercetin-Hex-2 (isoquercetrin)	16.20	C_21_H_20_O_12_	1.4	4.41 ± 0.65	2.74 ± 0.01	2
70	rhapontin	16.82	C_21_H_24_O_9_	2.0	3.46 ± 0.57	0.35 ± 0.11	1
75	quercetin-Pent-2 (avicularin)	17.51	C_20_H_18_O_11_	1.5	1.12 ± 0.20	ND	2
102	deoxyrhapontigenin-Hex-1	21.85	C_21_H_24_O_8_	0.4	1.29 ± 0.25	ND	3
103	(aloe-)emodin-anthrone-malonyl-Hex-1	22.09	C_24_H_24_O_12_	0.6	0.69 ± 0.10	0.37 ± 0.02	3
107	torachrysone-Hex-1	22.45	C_20_H_24_O_9_	0.2	1.42 ± 0.35	0.30 ± 0.04	3
109	(aloe-)emodin-dianthrone-di(malonyl-Hex)-1	22.85	C_48_H_46_O_24_	0.7	1.16 ± 0.26	0.36 ± 0.09	3
112	apigenin-7-Glu	23.02	C_21_H_20_O_10_	−0.3	0.85 ± 0.10	0.55 ± 0.11	1
113	pinocembrine-Hex-5	23.07	C_21_H_22_O_9_	−0.3	1.78 ± 0.22	0.41 ± 0.08	2
116	(aloe-)emodin-anthrone-malonyl-Hex-2	24.50	C_24_H_24_O_12_	−0.9	28.90 ± 2.80	4.32 ± 0.36	3
118	emodin-malonyl-Hex-2	24.55	C_24_H_22_O_13_	−0.9	ND	2.34 ± 0.20	3
121	torachrysone-Ac-Hex-2	24.79	C_22_H_26_O_10_	−0.3	6.51 ± 0.84	1.47 ± 0.15	3
123	nataloe-emodin-8-Me-Ac-Hex	25.50	C_24_H_26_O_11_	0.4	1.08 ± 0.11	0.43 ± 0.07	4
129	(aloe-)emodin-dianthrone-di(malonyl-Hex)-2	27.81	C_48_H_46_O_24_	1.2	2.72 ± 0.25	0.97 ± 0.11	3
130	physcion-anthrone-malonyl-Hex	27.92	C_25_H_26_O_12_	1.4	1.78 ± 0.22	ND	3
131	physcion-Ac-Hex-3	28.01	C_24_H_24_O_11_	0.7	1.67 ± 0.33	0.79 ± 0.11	3
133	(aloe-)emodin-dianthrone-di(malonyl-Hex)-3	28.70	C_48_H_46_O_24_	1.5	2.23 ± 0.28	0.84 ± 0.09	3
136	*Unidentified*	29.11	C_21_H_18_O_10_	0.4	1.36 ± 0.13	0.68 ± 0.08	4
137	emodin-dianthrone-malonyl-Hex-1	29.68	C_39_H_34_O_16_	1.1	1.91 ± 0.08	0.85 ± 0.14	3
140	emodin-dianthrone-malonyl-Hex-2	29.93	C_39_H_34_O_16_	0.3	1.60 ± 0.12	0.36 ± 0.05	3
143	emodin-dianthrone-malonyl-Hex-3	30.44	C_39_H_34_O_16_	−0.2	1.30 ± 0.09	0.86 ± 0.06	3
147	emodin-dianthrone-malonyl-Hex-4	30.64	C_39_H_34_O_16_	0.0	1.64 ± 0.19	0.74 ± 0.09	3
149	(aloe-)emodin-anthrone-2	30.85	C_15_H_12_O_4_	0.9	10.48 ± 1.80	1.98 ± 0.18	3
151	emodin	31.11	C_15_H_10_O_5_	0.9	4.77 ± 2.22	2.60 ± 0.14	1
155	emodin-dianthrone-1?	31.78	C_30_H_22_O_8_	0.5	1.20 ± 0.13	0.58 ± 0.10	3
159	emodin-dianthrone-2	32.03	C_30_H_22_O_8_	0.0	1.22 ± 0.06	0.43 ± 0.07	3
164	18:2-LPC-2	32.29	C_27_H_52_NO_9_P	0.5	1.69 ± 0.22	0.80 ± 0.06	2
166	18:2-LPE-2	32.33	C_23_H_44_NO_7_P	0.3	1.19 ± 0.15	0.67 ± 0.05	2
171	*Unidentified*	32.64	C_49_H_78_O_14_	−0.2	ND	0.84 ± 0.27	4
173	16:0-LPC-2	32.71	C_25_H_52_NO_9_P	0.4	0.92 ± 0.43	0.94 ± 0.30	2
174	16:0-LPE-2	32.71	C_21_H_44_NO_7_P	0.3	0.64 ± 0.30	0.68 ± 0.22	2
**No**	**Name**	**RT (min)**	**Calc. Formula**	**Err. (ppm)**	***R. rhabarbarum* Roots** **(mg/g d.w.)**	***R. rhaponticum* Roots** **(mg/g d.w.)**	**ID Level**
1	Hex-Hex	1.12	C_12_H_22_O_11_	−0.4	23.7 ± 1.58	110.9 ± 6.11	3
2	malate	1.28	C_4_H_6_O_5_	1.5	4.6 ± 0.44	ND	2
3	citrate	1.43	C_6_H_8_O_7_	0.2	0.8 ± 0.15	1.5 ± 0.11	2
4	tyrosine	1.68	C_9_H_11_NO_3_	1.7	0.6 ± 0.03	0.8 ± 0.12	1
5	beta-glucogallin	1.92	C_13_H_16_O_10_	0.7	31.2 ± 2.39	6.4 ± 0.23	1
6	beta-glucogallin-2	2.21	C_13_H_16_O_10_	1.2	1.0 ± 0.14	3.0 ± 0.28	3
7	gallate	2.41	C_7_H_6_O_5_	4.4	1.2 ± 0.24	ND	1
8	beta-glucogallin-3	2.44	C_13_H_16_O_10_	0.9	1.4 ± 0.11	4.5 ± 0.35	3
17	Trp	5.74	C_11_H_12_N_2_O_2_	0.1	0.9 ± 0.11	0.8 ± 0.05	1
28	digalloyl-Hex-5	7.60	C_20_H_20_O_14_	−0.9	6.1 ± 0.43	8.5 ± 0.46	3
30	catechin	7.83	C_15_H_14_O_6_	−0.9	10.1 ± 0.84	1.8 ± 0.11	1
31	*Unidentified*	7.98	C_21_H_22_O_11_	−0.8	ND	1.2 ± 0.08	4
32	*Unidentified*	8.05	C_17_H_30_O_13_	1.8	1.2 ± 0.29	ND	4
33	digalloyl-Hex-7	8.08	C_20_H_20_O_14_	1.9	3.1 ± 0.30	1.5 ± 0.07	3
34	coumaroyl-Hex-3	8.15	C_15_H_18_O_8_	−2.1	2.1 ± 0.28	ND	3
36	resveratrol-diHex-1	8.61	C_26_H_32_O_13_	1.2	0.6 ± 0.17	0.7 ± 0.03	3
39	digalloyl-Hex-8	8.86	C_20_H_20_O_14_	−0.8	ND	0.7 ± 0.06	3
40	benzoyl-Hex-Pent	8.93	C_18_H_24_O_11_	2,0	1.5 ± 0.05	ND	3
42	procyanidin-B2	9.25	C_30_H_26_O_12_	−0.8	ND	1.3 ± 0.18	1
43	hydroxybenozyl-galloyl-Hex-1	9.27	C_20_H_20_O_12_	−1.8	ND	0.7 ± 0.08	3
46	*Unidentified*	9.44	C_17_H_24_O_9_	2.3	1.1 ± 0.24	ND	4
51	piceatannol/oxyresveratrol-diHex-2	10.02	C_26_H_32_O_14_	2.0	0.9 ± 0.09	1.3 ± 0.02	3
61	(epi-)catechin-(epi-)catechin-gallate-1	10.71	C_37_H_30_O_16_	−1.4	1.1 ± 0.17	ND	3
62	*Unidentified*	10.83	C_35_H_34_O_15_	−0.5	ND	1.2 ± 0.10	4
64	piceatannol-Hex-1	11.12	C_20_H_22_O_9_	1.0	6.7 ± 0.55	4.2 ± 0.27	3
66	piceatannol-Hex-2	11.30	C_20_H_22_O_9_	−0.4	1.8 ± 0.26	2.7 ± 0.24	3
69	resveratrol-Hex-1 (resveratroloside)	11.62	C_20_H_22_O_8_	0.1	24.5 ± 1.98	21.1 ± 1.14	2
74	hydroxybenozyl-galloyl-Hex-4	11.87	C_20_H_20_O_12_	−1.4	ND	0.7 ± 0.06	3
75	resveratrol-diHex-2	11.91	C_26_H_32_O_13_	−0.1	0.5 ± 0.06	1.6 ± 0.27	3
78	*Unidentified*	12.04	C_29_H_34_O_17_	0.1	ND	1.3 ± 0.14	4
79	trihydroxyresveratrol	12.07	C_14_H_12_O_6_	−0.6	ND	1.1 ± 0.18	3
84	astringin	12.68	C_20_H_22_O_9_	0.4	65.3 ± 4.38	99.3 ± 4.20	1
88	*Unidentified*	13.06	C_21_H_28_O_13_	−1.2	5.1 ± 0.79	4.9 ± 0.44	4
89	polyflavanostilbene-A	13.10	C_42_H_38_O_19_	−0.4	4.7 ± 0.45	ND	3
90	digalloyl-procyanidin-B2-1	13.12	C_44_H_34_O_20_	0.4	ND	3.8 ± 0.69	3
91	polydatin	13.20	C_20_H_22_O_8_	−0.1	14.0 ± 0.72	11.3 ± 0.56	1
96	piceatannol-galloyl-Hex-1	13.35	C_27_H_26_O_13_	−0.1	1.7 ± 0.25	ND	3
99	galloyl-catechin-2	13.47	C_22_H_18_O_10_	−0.2	6.7 ± 0.52	8.0 ± 0.36	3
108	*Unidentified*	13.91	C_43_H_40_O_19_	−0.5	1.7 ± 0.08	ND	4
110	piceatannol-Pent	14.01	C_19_H_20_O_8_	−0.4	3.4 ± 0.44	4.7 ± 0.50	3
112	piceatannol-galloyl-Hex-2	14.10	C_27_H_26_O_13_	−0.4	4.4 ± 1.09	11.3 ± 0.78	3
113	(iso-)rhapontigenin-Hex-1	14.20	C_21_H_24_O_9_	−0.4	18.5 ± 0.55	17.7 ± 0.73	3
115	pinocembrine-Hex-2	14.34	C_21_H_22_O_9_	−1.2	ND	1.6 ± 0.11	3
116	trans-piceatannol	14.43	C_14_H_12_O_4_	−0.5	2.5 ± 0.50	4.0 ± 1.17	1
121	rhapontin	14.84	C_21_H_24_O_9_	−0.3	184.0 ± 10.93	151.3 ± 7.77	1
129	(iso-)rhapontigenin-galloyl-Hex-1	15.32	C_28_H_28_O_13_	0.1	3.8 ± 0.70	5.0 ± 1.77	3
134	eriodictyol-Hex	15.79	C_21_H_22_O_11_	0.1	1.0 ± 0.18	2.6 ± 0.28	3
140	(iso-)rhapontigenin-galloyl-Hex-2	16.21	C_28_H_28_O_13_	1.1	13.3 ± 0.58	9.4 ± 0.74	3
142	(iso-)rhapontigenin-malonyl-Hex-1	16.40	C_24_H_26_O_12_	−0.1	11.2 ± 1.09	5.7 ± 0.41	3
145	piceatannol-hydroxybenzoyl-Hex	16.66	C_27_H_26_O_11_	0.7	0.6 ± 0.11	ND	3
147	emodin-Hex-3	16.71	C_21_H_20_O_10_	1.1	1.2 ± 0.20	12.8 ± 0.68	3
158	*Unidentified*	17.31	C_42_H_34_O_9_	1.2	6.7 ± 0.59	5.6 ± 0.49	4
161	*Unidentified*	17.49	C_42_H_34_O_9_	1.1	3.5 ± 0.13	2.3 ± 0.13	4
163	piceatannol-dimer (cararosinol-D)	17.55	C_28_H_22_O_8_	1.5	ND	1.5 ± 0.40	2
167	piceatannol-coumaroyl-Hex-2	17.68	C_29_H_28_O_11_	1.0	3.9 ± 0.36	5.1 ± 0.24	3
168	*Unidentified*	17.77	C_24_H_28_O_11_	1.5	1.2 ± 0.32	ND	4
169	piceatannol-feruloyl-Hex	17.83	C_30_H_30_O_12_	1.0	0.6 ± 0.08	3.8 ± 0.24	3
172	(aloe-)emodin-galloyl-Hex	18.07	C_28_H_24_O_14_	0.9	ND	2.0 ± 0.30	3
180	(iso-)rhapontigenin	18.56	C_15_H_14_O_4_	2.3	12.6 ± 0.79	10.7 ± 0.85	3
193	deoxyrhapontigenin-Hex-1	19.18	C_21_H_24_O_8_	1.7	138.6 ± 8.55	31.7 ± 2.08	3
199	chrysophanol-Hex-1	19.39	C_21_H_20_O_9_	1.2	4.2 ± 0.82	17.4 ± 0.91	3
201	apigenin-7-Glu	19.51	C_21_H_20_O_10_	1.0	1.1 ± 0.17	3.5 ± 0.19	1
203	(iso-)rhapontigenin-coumaroyl-Hex-2	19.67	C_30_H_30_O_11_	0.9	3.9 ± 0.60	3.0 ± 0.29	3
205	chrysophanol-Hex-2	19.81	C_21_H_20_O_9_	1.3	2.5 ± 0.26	12.5 ± 0.73	3
208	(iso-)rhapontigenin-feruloyl-Hex-2	19.93	C_31_H_32_O_12_	1.0	ND	2.0 ± 0.40	3
209	deoxyrhapontigenin-galloyl-Hex-1	20,00	C_28_H_28_O_12_	1.4	6.8 ± 0.74	1.8 ± 0.22	3
210	flavanol-piceatannol-dimer	20.01	C_29_H_24_O_8_	1.6	ND	3.4 ± 0.43	3
212	rhapontigenin-coumaroyl-Hex-2	20.42	C_30_H_30_O_11_	1.7	ND	2.1 ± 0.13	3
214	deoxyrhapontigenin-malonyl-Hex	20.44	C_24_H_26_O_11_	1.4	3.0 ± 0.31	1.2 ± 0.13	3
218	rhapontigenin-feruloyl-Hex-3	20.72	C_31_H_32_O_12_	1.6	ND	0.7 ± 0.13	3
221	(aloe-)emodin-anthrone-malonyl-Hex(-CO2)-2	20.94	C_23_H_24_O_10_	1.3	2.2 ± 0.16	ND	3
222	resveratrol-dimer-1	20.95	C_28_H_22_O_6_	1.4	3.3 ± 0.29	3.4 ± 0.16	3
223	torachrysone-malonyl-Hex(-CO2)-3	21.08	C_22_H_26_O_10_	1.4	5.1 ± 0.46	5.0 ± 0.29	3
227	physcion-Hex-1 (rheochrysin)	21.24	C_21_H_20_O_8_	0.1	ND	2.2 ± 0.38	3
229	resveratrol-piceatannol-mixed-tetramer	21.33	C_38_H_52_O_25_	−2.0	0.8 ± 0.07	2.9 ± 0.59	3
231	chrysophanol-Ac-Hex-3	21.45	C_23_H_22_O_10_	2.7	6.0 ± 0.52	ND	3
232	chrysophanol-physcion-dianthr.-di(malonyl-Hex)-1	21.45	C_48_H_44_O_24_	1.0	ND	16.0 ± 0.72	3
240	chrysophanol-malonyl-Hex(-CO2)-4	21.82	C_23_H_22_O_10_	0.2	5.8 ± 0.36	ND	3
241	*Unidentified*	21.82	C_46_H_44_O_22_	0.8	ND	4.7 ± 1.40	4
242	chrysophanol-physcion-dianthr.-di(malonyl-Hex)-2	21.82	C_48_H_44_O_24_	1.7	ND	11.5 ± 1.15	3
252	chrysophanol-dianthrone-di(malonyl-Hex)-2	22.49	C_48_H_46_O_22_	1.5	1.0 ± 0.13	ND	3
258	resveratrol-dimer-2	23.10	C_28_H_22_O_6_	2.7	8.4 ± 0.61	9.4 ± 0.57	3
267	chrysophanol-anthrone-malonyl-Hex-4	23.45	C_24_H_24_O_11_	−0.6	ND	4.4 ± 0.33	3
272	chrysophanol-dianthrone-malonyl-diHex-8	23.56	C_45_H_44_O_19_	1.9	0.7 ± 0.08	ND	3
278	deoxyrhapontigenin	23.89	C_15_H_14_O_3_	3.6	13.4 ± 1.16	2.2 ± 0.21	1
286	chrysophanol-dianthrone-di(malonyl-Hex)-6	24.25	C_48_H_46_O_22_	1.2	0.8 ± 0.11	ND	3
291	chrysophanol-dianthrone-di(malonyl-Hex)-7	24.47	C_48_H_46_O_22_	1.8	1.5 ± 0.25	ND	3
309	chrysophanol-dianthrone-di(malonyl-Hex)-9	26.55	C_48_H_46_O_22_	1.7	0.9 ± 0.03	ND	3
322	emodin	29.75	C_15_H_10_O_5_	0.9	0.8 ± 0.12	2.8 ± 0.11	1
324	chrysophanol-dianthr.-malonyl-Hex-2	29.91	C_39_H_34_O_14_	−0.9	3.2 ± 0.26	ND	3
327	chrysophanol-physcion-dianthr.-malonyl-Hex-2	30.11	C_40_H_36_O_15_	−0.7	1.2 ± 0.14	ND	3

**Table 3 nutrients-15-00949-t003:** The in vitro assessment of cytotoxicity of the examined rhubarb extracts and stilbenes towards HUVECs. Data are presented as mean ± SD, *n* = 4. The viability of control (untreated) HUVECs was assumed as 100%.

Extract/Compounds	Concentration (µg/mL)	% of Cell Viability ± SD
*R. rhaponticum* petiole extract	1	102.94 ± 4.62
	5	96.24 ± 4.55
	50	88.81 ± 2.83
*R. rhabarbarum* petiole extract	1	101.94 ± 5.61
	5	97.38 ± 6.54
	50	98.52 ± 7.58
*R. rhaponticum* root extract	1	104.99 ± 4.97
	5	105.43 ± 4.49
	50	89.54 ± 5.47
*R. rhabarbarum* root extract	1	97.34 ± 4.43
	5	97.86 ± 6.95
	30	83.45 ± 7.06
Rhapontigenin	1	98.59 ± 14.9
	5	103.12 ± 12.10
	25	88.32 ± 7.75
Rhaponticin	1	103.38 ± 18.84
	5	106.72 ± 12.70
	50	109.32 ± 17.01
Indomethacin	5	95.83 ± 5.63
Zileuton	5	105.28 ± 3.56

**Table 4 nutrients-15-00949-t004:** Computational analysis of the most abundant compounds found in *R. rhaponticum* and *R. rhabarbarum*, native substrate native arachidonic acid and inhibitors: indomethacin and NDGA: MW—predicted molecular weight, PSA—molecular polar surface area, HA—the number of heavy atoms, MBS PI—protease inhibitor Molinspiration bioactivity score v2014.03, MBS EI—enzyme inhibitor Molinspiration bioactivity score v2014.03, ΔG°—molecular docking predicted standard free energy change in ligand binding—values for COX-2 up, 5-LOX bottom, respectively, LE—ligand efficiency (LE = −RTlnKd/HA or −ΔG°/HA), LELP = milog P/LE.

(No.)	Compound Names, Chemical Structure and SMILES	MW (Da) PSA	HA	milogP	MBS PI	MBS EI	ΔG°_bind_ (kcal·mol^−1^) COX-2 5-LOX	LELELP
(1)	digalloyl glucoside 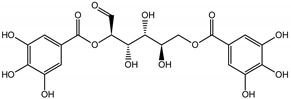 c1c(C(=O)OC[C@H]([C@H]([C@H]([C@H](OC(=O)c2cc(c(c(O)c2)O)O)C=O)O)O)O)cc(c(c1O)O)O	484.37 251.73	34	−0.31	0.01	0.13	−8.5 ± 0.2 −1.5 ± 1.6	0.25 −1.24 0.04 −7.75
(2)	glucogallin 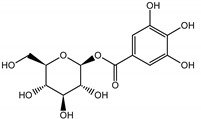 c1(cc(cc(c1O)O)C(=O)O[C@H]1[C@@H]([C@H]([C@@H]([C@H](O1)CO)O)O)O)O	332.26 177.13	23	−1.48	0.07	0.42	−7.7 ± 0.1 −5.9 ± 1.0	0.33 −4.48 0.26 −5.69
(3)	syringoyl 1-*O*-glucopyranoside 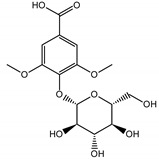 [C@@H]1([C@@H]([C@H]([C@@H](O[C@@H]1CO)Oc1c(cc(cc1OC)C(=O)O)OC)O)O)O	360.31 155.15	25	−0.84	−0.02	0.30	−7.0 ± 0.2 −2.8 ± 0.6	0.28 −3.00 0.11 −7.64
(4)	astringin 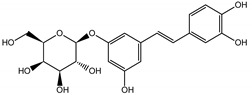 c1c(O)cc(cc1/C=C/c1cc(c(cc1)O)O)O[C@H]1[C@@H]([C@H]([C@H]([C@H](O1)CO)O)O)O	406.39 160.06	29	0.71	0.04	0.34	−8.1 ± 0.1 0.4 ± 1.4	0.28 2.54 −0.01 −71.00
(5)	piceatannol-galloylglucoside 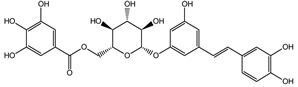 c1c(c(cc(c1)/C=C/c1cc(cc(c1)O)O[C@H]1[C@@H]([C@H]([C@@H]([C@H](O1)COC(=O)c1cc(c(c(c1)O)O)O)O)O)O)O)O	558.49 226.82	40	1.88	−0.06	0.12	−9.3 ± 0.2 3.5 ± 2.7	0.23 8.17 −0.09 −20.89
(6)	piceid (polydatin; resveratrol 3-*O*-beta-glucopyranoside) 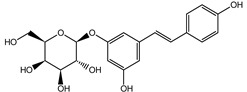 C1=CC(=CC=C1/C=C/C2=CC(=CC(=C2)O[C@H]3[C@@H]([C@H]([C@@H]([C@H](O3)CO)O)O)O)O)O	390.39 139.84	28	1.20	0.05	0.34	−8.5 ± 0.2 0.5 ± 1.6	0.30 4.00 −0.018 −66.67
(7)	viniferin (resveratrol-dehydrodimer) 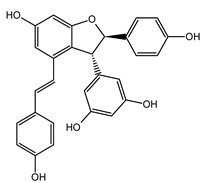 c1cc(ccc1/C=C/c1cc(cc2c1[C@H]([C@@H](O2)c1ccc(cc1)O)c1cc(cc(c1)O)O)O)O	454.48 110.37	34	4.77	−0.4	0.20	−8.8 ± 0.1 7.0 ± 0.1	0.26 18.35 −0.21 −22.71
(8)	deoxyrhaponticin 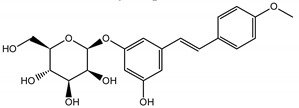 c1c(ccc(c1)/C=C/c1cc(cc(c1)O[C@H]1[C@H]([C@H]([C@@H]([C@H](O1)CO)O)O)O)O)OC	404.42 128.84	26	1.74	0.01	0.30	−8.8 ± 0.1 1.2 ± 1.5	0.34 5.12 −0.05 −34.8
(9)	deoxyrhapontigenin 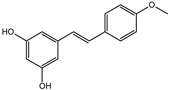	243.27 49.69	18	3.52	−0.40	−0.01	−7.6 ± 0.1 −5.2 ± 0.1	0.42 8.38 0.29 12.14
(10)	rhaponticin 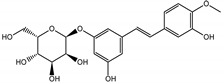 c1c(cc(cc1O[C@H]1[C@H]([C@H]([C@@H]([C@@H](O1)CO)O)O)O)/C=C/c1ccc(c(c1)O)OC)O	420.41 149.07	30	1.02	−0.02	0.30	−8.5 ± 0.2 0.4 ± 3.5	0.29 42.57 −0.01 −102.0
(11)	rhapontigenin 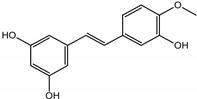 c1c(c(cc(c1)/C=C/c1cc(cc(c1)O)O)O)O	244.25 80.91	18	2.50	−0.34	0.07	−7.7 ± 0.1 −5.6 ± 0.2	0.43 5.81 0.31 8.06
(12)	rhapontigenin-galloyl-glucopyranoside 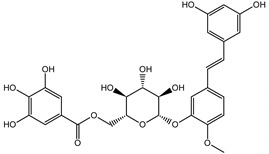 c1c(c(cc(c1)/C=C/c1cc(cc(c1)O)O)O[C@H]1[C@@H]([C@H]([C@@H]([C@H](O1)COC(=O)c1cc(c(c(c1)O)O)O)O)O)O)OC	572.52 215.83	41	1.96	−0.10	0.05	−7.8 ± 0.2 5.60 ± 3.1	0.19 10.32 −0.14 −14
(13)	(epi)catechin 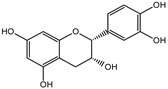 c1(cc(c2c(c1)O[C@@H]([C@@H](C2)O)c1cc(c(cc1)O)O)O)O	290.27 110.37	21	1.37	0.26	0.47	−8.1 ± 0.1 −4.3 ± 0.1	0.39 3.51 0.21 6.2
(14)	isoquercitrin 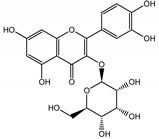 c1(cc(c2c(c1)[OH+]C(=C(C2=O)O[C@@H]1[C@@H]([C@@H]([C@@H]([C@H](O1)CO)O)O)O)c1ccc(c(c1)O)O)O)O	465.39 211.69	33	−3.57	−0.04	0.29	−7.0 ± 0.2 4.5 ± 2.0	0.21 −17.0 −0.14 25.5
(15)	rutin 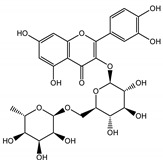 C[C@H]1[C@@H]([C@H]([C@H]([C@@H](O1)OC[C@@H]2[C@H]([C@@H]([C@H]([C@@H](O2)OC3=C(OC4=CC(=CC(=C4C3=O)O)O)C5=CC(=C(C=C5)O)O)O)O)O)O)O)O	610.52 269.43	43	−1.06	−0.07	0.12	−7.9 ± 0.1 11.3 ± 1.5	0.18 −5.09 −0.26 4.08
(16)	emodin 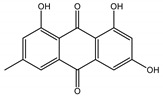 c1c(cc2c(c1O)C(=O)c1c(C2=O)cc(cc1O)C)O	270.24 94.83	20	3.01	−0.21	0.21	−8.9 ± 0.1 −6.4 ± 0.1	0.45 6.67 0.32 9.41
(17)	emodin 8-*O*-glucoside 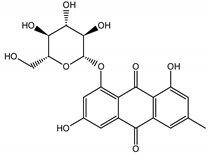 [C@@H]1([C@H](Oc2c3c(C(=O)c4c(C3=O)c(cc(c4)C)O)cc(c2)O)O[C@@H]([C@H]([C@@H]1O)O)CO)O	432.38 173.98	31	0.96	0.05	0.41	−6.9 ± 0.3 0.7 ± 0.8	0.22 4.36 −0.02 −48.00
(18)	emodin anthrone 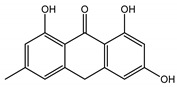 c1c(cc2c(c1O)C(=O)c1c(C2)cc(cc1O)C)O	256.26 77.75	19	3.25	−0.29	0.30	−8.8 ± 0.0 −7.2 ± 0.2	0.46 7.06 0.38 8.55
(19)	chrysophanol-8-glucoside 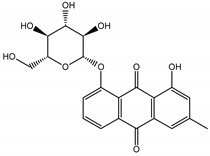 c1ccc2c(c1O[C@H]1[C@@H]([C@H]([C@@H]([C@H](O1)CO)O)O)O)C(=O)c1c(C2=O)cc(cc1O)C	416.38 153.75	30	1.49	0.06	0.39	−8.4 ± 0.0 0.7 ± 3.2	0.28 5.32 −0.02 −74.50
(20)	arachidonic acid 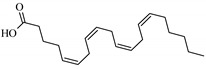 OC(=O)CCC/C=C\C/C=C\C/C=C\C/C=C\CCCCC	304.47 37.30	22	6.42	0.19	0.35	−7.4 ± 0.2 −5.1 ± 0.3	0.34 18.88 0.23 27.91
(21)	NDGA 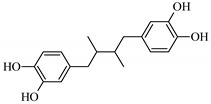 CC(CC1=CC(=C(C=C1)O)O)C(C)CC2=CC(=C(C=C2)O)O	302.37 80.91	22	3.48	0.01	0.13	−8.0 ± 0.2 −5.2 ± 0.5	0.34 10.24 0.035 99.43
(22)	indomethacin 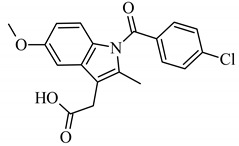 C(=O)(n1c(C)c(c2cc(ccc12)OC)CC(=O)O)c1ccc(Cl)cc1	357.79 68.64	25	3.99	−0.11	0.30	−9.4 ± 1.3 0.8 ± 1.3	0.38 10.50 −0.03 −133.00

## Data Availability

Not applicable.

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
