# Peer review of "Rheum rhaponticum and Rheum rhabarbarum Extracts as Modulators of Endothelial Cell Inflammatory Response"

_nutrients, 2023, doi:10.3390/nu15040949_

Round 1

Reviewer 1 Report

The present study evaluates the anti-inflammatory potential of rhubarb extracts, isolated from petioles and underground organs of Rheum rhabarbarum L. (garden rhubarb) and R. rhaponticum L. (rhapontic rhubarb) as well as rhapontigenin (RHPG) and its glycoside, rhaponticin (RHPT). By detecting release of the inflammatory mediators, cyclooxygenase (COX-2) and 5-lipoxygenase (5-LOX) expression as well as recruitment of leukocytes to the activated  endothelial cells (HUVECs), authors revealed significant difference in the activities of preparations from the petioles and root extracts. In silico studies also indicated that organic chemical compound in the extract bind to COX-2, instead of 5-LOX. The comprehensive study intensified the knowledge of these herbs. There are mistakes that needed to be revised.

Line111: “with methanol containing 0,1 % formic acid” 0,1 % should be 0.1 % or not ?

Line190:  “at a density of 10x103 cells/well.” 10x103 cells should be 1x104.

Line343: “(Table 2S, Figure 2S, Figure 4S, Figure 1)” Full stop symbol missing.

Line394: “semi-quantitation” should “semi” be italic, please make it consistent in the manuscript.

Line420: “(Figure 1 E, ...)” should be Figure 2E.

Line436: “4xmagnification” should be 4x magnification.

Line444: Data for “% of control ± SD” should be revised.

Line570: “at 3 µg/ml concentration.” Font size non-consistent.

Text in figures was illegible, please provide clear illustrations.

Author Response

The present study evaluates the anti-inflammatory potential of rhubarb extracts, isolated from
petioles and underground organs of Rheum rhabarbarum L. (garden rhubarb) and R.
rhaponticum L. (rhapontic rhubarb) as well as rhapontigenin (RHPG) and its glycoside,
rhaponticin (RHPT). By detecting release of the inflammatory mediators, cyclooxygenase
(COX-2) and 5-lipoxygenase (5-LOX) expression as well as recruitment of leukocytes to the
activated endothelial cells (HUVECs), authors revealed significant difference in the activities
of preparations from the petioles and root extracts. In silico studies also indicated that organic
chemical compound in the extract bind to COX-2, instead of 5-LOX. The comprehensive study
intensified the knowledge of these herbs. There are mistakes that needed to be revised.
The authors would like to thank the Reviewer for valuable comments and suggestions.
Following directions of the Reviewer, the manuscript text has been improved.
Line111: “with methanol containing 0,1 % formic acid” 0,1 % should be 0.1 % or not ?
The concentration should be indicated as 0.1 %. The mistake has been corrected.
Line190: “at a density of 10x103 cells/well.” 10x103 cells should be 1x104.
Information on cell count has been corrected.
Line343: “(Table 2S, Figure 2S, Figure 4S, Figure 1)” Full stop symbol missing.
The missing full stops have been added into the indicated text.
Line394: “semi-quantitation” should “semi” be italic, please make it consistent in the
manuscript.
According to suggestion of the Reviewer, italics have been added in the manuscript.
Line420: “(Figure 1 E, ...)” should be Figure 2E.
The mistake has been corrected.
Line436: “4xmagnification” should be 4x magnification.
The statement concerning magnification has been corrected.
Line444: Data for “% of control ± SD” should be revised.
Information on the control has been corrected according to suggestions of the Reviewer.
Line570: “at 3 µg/ml concentration.” Font size non-consistent.
The size of above text fragment has been corrected.
Text in figures was illegible, please provide clear illustrations.
Low quality of the figures was probably a result of a conversion of the image files into the *.pdf
file format of the manuscript. In the revised version of the manuscript, we have added images
in a higher resolution.

Reviewer 2 Report

The manuscript “Rheum rhaponticum and Rheum rhabarbarum extracts as modulators of endothelial cell inflammatory response” is an interesting work that can lead to more inquisitive and complete projects. There are some questions and concerns regarding this work.

-          What are the criteria for determining the cellular safety in viability test?

-          List of compounds in phytochemical profile analysis described in the Results section can be summarize using table and shown in the main text.

-          The common and unique metabolites from each part of the plant in the same species are interesting to be observed and presented.

-          Name of the species in figure 5-7 should be italic.

-          Results 3.4 and 3.5 were confusingly described in the text and not correlated with the data shown in the representative figures. For example, in Results 3.4, the authors mentioned that “Extract from the petioles of R. rhaponticum and R. rhabarbarum did not influence the COX-2 gene expression (Figure 5A) in unstimulated HUVECs, while in the LPS-activated cells (Figure 5B), a statistically significant (***p < 0.001) reduction of COX-2 mRNA level was found.”, however the petiole extracts from both species showed the significant difference in Figure 5A. Moreover, the authors should extensively describe “the statistically significant differences”, what group used for comparing to the extract/compound groups.

-          The authors only investigated the inflammatory responses of the endothelial cells, however it is still unclear that what is the benefit of these responses to the cells? Can the extracts recuse the cells from inflammation-induced cytotoxicity?

Author Response

The manuscript “Rheum rhaponticum and Rheum rhabarbarum extracts as modulators of
endothelial cell inflammatory response” is an interesting work that can lead to more inquisitive
and complete projects. There are some questions and concerns regarding this work.
The authors would like to thank the Reviewer for valuable opinions and suggestions. Following
directions of the Reviewer, the manuscript text has been improved.
What are the criteria for determining the cellular safety in viability test?
Missing information has been added into the methodological section of the revised manuscript
(into the Chapter 2.4.) Cell viability was estimated during the metabolic, resazurin-based test.
The metabolic activity of control HUVECs (untreated with the examined extracts) was assumed
as 100% of viability. Samples treated with 1% Triton-X100 were reference samples, with no
viable cells (0% of viability). In cell samples treated with the examined extracts or stilbenes, a
decrease of cell viability 20% (compared to control/untreated HUVECs) was assumed as a
cytotoxic effect.
List of compounds in phytochemical profile analysis described in the Results section can be
summarize using table and shown in the main text.
The manuscript is primarily devoted to the biological activity of the examined plantderived extracts, and results from phytochemical analyses are supporting data. The number
of detected substances exceeded 150 and 300 compounds in the petioles and roots,
respectively. During preparation of the original version of the manuscript, we preliminary
constructed tables with full characteristics of the examined extracts - however, in the MDPI
Nutrients template format, these phytochemical tables covered over 36 pages. For that reason
a detailed characteristics of the examined extracts (including tables and UHPLC profiles) has
been fully presented in the Supplementary Materials (i.e. into the Supplementary materials 1
file, the Supplementary_Table 1S and the Supplementary_Table 2S), attached into the
manuscript.
Following suggestion of the Reviewer, in the revised version of the manuscript, we have
included an additional table (Table 2) to the main text, to present the main metabolites in all
the investigated tissues of both species. We hope that the incorporation of the Table 2 into the
main text will make our manuscript more readable. However, most of these data have been
already extensively presented in the Supplementary Materials.
The common and unique metabolites from each part of the plant in the same species are
interesting to be observed and presented.
Additionally, during revision of the manuscript, we prepared the Table 3S, which summarizes
data used to prepare Figure 1 and contains the common and unique metabolites from each
part of the plant in the same species. Since this table content partly duplicates information
already given in the Table 2 (the main body of the manuscript), it has been included into the
Supplementary Materials, and assigned by the name: Supplementary_Table 3S.
Name of the species in figure 5-7 should be italic.
Following directions of the Reviewer, italics have been used in the figures.
2
Results 3.4 and 3.5 were confusingly described in the text and not correlated with the data
shown in the representative figures. For example, in Results 3.4, the authors mentioned that
“Extract from the petioles of R. rhaponticum and R. rhabarbarum did not influence the COX-2
gene expression (Figure 5A) in unstimulated HUVECs, while in the LPS-activated cells (Figure
5B), a statistically significant (***p < 0.001) reduction of COX-2 mRNA level was found.”,
however the petiole extracts from both species showed the significant difference in Figure 5A.
Moreover, the authors should extensively describe “the statistically significant differences”,
what group used for comparing to the extract/compound groups.
Following suggestions of the Reviewer the text has been revised in order to more precisely
described the results. Furthermore, at the beginning of the 3.3. Evaluation of the antiinflammatory properties of the examined substances chapter, an introducing fragment has
been added, to explain the used experimental models and control samples:
Anti-inflammatory effects of the examined extracts and stilbenes were monitored at a
cytophysiological level (cytokine release from HUVECs and interactions with monocytes) as
well as at a molecular level of intracellular processes (the expression of pro-inflammatory
enzyme genes).
To evaluate effects of the examined substances on cytokine secretion and interactions
with monocytes, HUVECs were pre-incubated with the extracts or stilbenes, and then
stimulated with LPS. In these assays, anti-inflammatory efficiency of the examined extracts
and stilbenes was estimated by comparing the cytokine level or monocyte influx in HUVECs
samples preincubated with the plant substances and activated by LPS, to cells stimulated with
LPS in the absence of the examined extracts and stilbenes.
Analyses of the COX-2 and 5-LOX gene expression required the use of two
experimental models. The plant-derived substances are exogenous factors that may
themselves influence gene expression, as a part of the cell adaptive response. Therefore, in
the first one of the used experimental models, the expression of COX-2 and 5-LOX genes was
studied in HUVECS pre-incubated with the examined plant substances, without subsequent
stimulation with LPS. This assay enabled verification if the examined extracts influenced the
HUVECs at their physiological state (under physiological conditions, with no exposure to proinflammatory stimuli). Effects of the rhubarb extracts and stilbenes on COX-2 and 5-LOX gene
expression were evaluated by comparing with control samples, i.e. native HUVECs (untreated
by these plant substances or LPS).
The second experimental model was designed to study anti-inflammatory properties of
rhubarb extracts and stilbenes under inflammatory conditions (i.e. in the LPS-stimulated cells).
HUVECS were pre-incubated with the extracts or stilbenes, and then stimulated with LPS. The
anti-inflammatory action of the examined substances was evaluated by comparison of the
COX-2 and 5-LOX genes expression in these samples to the gene expression in HUVECs
treated with the LPS in the absence of the rhubarb extracts or stilbenes.
Additionally, the subchapter numbering has been modified to make the chapter 3 more
consistent. The subchapter 3.4. was replaced by 3.3.2, and so on. Since all of them present
results from evaluation of anti-inflammatory effects in HUVECs, their presentation in a common
chapter 3.3. will be more clear.
The authors only investigated the inflammatory responses of the endothelial cells, however it
is still unclear that what is the benefit of these responses to the cells? Can the extracts recuse
the cells from inflammation-induced cytotoxicity?
3
Following suggestions of the Reviewer, we revised the Discussion section and added missing
information.

Round 2

Reviewer 1 Report

Accept the revised version

Reviewer 2 Report

-